# A Toll-receptor map underlies structural brain plasticity

**Guiyi Li[1,2], Manuel G Forero[3], Jill S Wentzell[1†], Ilgim Durmus[1], Reinhard Wolf[2], Niki C Anthoney[1‡], Mieczyslaw Parker[1], Ruiying Jiang[1], Jacob Hasenauer[1], Nicholas James Strausfeld[2,4], Martin Heisenberg[2], Alicia Hidalgo[1]***

[1]Neurodevelopment Lab, School of Biosciences, University of Birmingham, Birmingham, United Kingdom; [2]Rudolf Virchow Center for Experimental Biomedicine, University of Würzburg, Würzburg, Germany; [3]Facultad de Ingeniería, Universidad de Ibagué, Ibagué, Colombia; [4]Neuroscience, University of Arizona College of Science, Tucson, United States

**\*For correspondence:**
a.hidalgo@bham.ac.uk

**Present address:**
[†]Undergraduate Advising and Research, Stanford University, Stanford, United States;
[‡]Medical School, University of Birmingham, Birmingham, United Kingdom

**Competing interests:** The authors declare that no competing interests exist.

**Abstract** Experience alters brain structure, but the underlying mechanism remained unknown. Structural plasticity reveals that brain function is encoded in generative changes to cells that compete with destructive processes driving neurodegeneration. At an adult critical period, experience increases fiber number and brain size in *Drosophila*. Here, we asked if Toll receptors are involved. Tolls demarcate a map of brain anatomical domains. Focusing on *Toll-2,* loss of function caused apoptosis, neurite atrophy and impaired behaviour. Toll-2 gain of function and neuronal activity at the critical period increased cell number. Toll-2 induced cycling of adult progenitor cells via a novel pathway, that antagonized MyD88-dependent quiescence, and engaged Weckle and Yorkie downstream. Constant knock-down of multiple *Tolls* synergistically reduced brain size. Conditional over-expression of *Toll-2* and *wek* at the adult critical period increased brain size. Through their topographic distribution, Toll receptors regulate neuronal number and brain size, modulating structural plasticity in the adult brain.

## Introduction

Structural brain plasticity and neurodegeneration reveal generative and destructive processes operating in the brain. Plasticity reflects adaptations of the brain to environmental change, involving adult neurogenesis, growth of neurites and synapses, which correlate with learning, experience, physical exercise and anti-depressant treatment (*Holtmaat and Svoboda, 2009*; *Deng et al., 2010*); conversely, neuroinflammation, neurodegeneration, loss of neurons, neurites and synapses, correlate with ageing, stress, depression and disease (*Wohleb et al., 2016*). Structural brain plasticity affects the brain topographically, influencing the specific regions involved in experience-dependent processing. These manifestations suggest that brain function is encoded in physical changes to cells.

Structural plasticity occurs in the *Drosophila* brain (*Sugie et al., 2018*). Breeding adult flies in constant darkness decreases, and in constant light increases brain volume (*Barth and Heisenberg, 1997*; *Barth et al., 1997*). Breeding adult flies in isolation vs. crowded conditions, or in single sex vs. mixed groups, also causes brain volume changes (*Technau, 1984*; *Heisenberg et al., 1995*). The affected modules include the optic lobe, the mushroom body calyx and central complex (*Technau, 1984*; *Heisenberg et al., 1995*; *Barth and Heisenberg, 1997*; *Barth et al., 1997*). Changes in brain volume are prominent in a critical period spanning from adult eclosion to day 5, and correlate with changes in fiber number (*Technau, 1984*; *Heisenberg et al., 1995*; *Barth and Heisenberg, 1997*; *Barth et al., 1997*). The molecular mechanisms underlying structural brain plasticity are unknown, and discovering them is crucial to understand the normal functionality of the brain as well as its pathological responses to disease.

**eLife digest** Everything that you experience leaves its mark on your brain. When you learn something new, the neurons involved in the learning episode grow new projections and form new connections. Your brain may even produce new neurons. Physical exercise can induce similar changes, as can taking antidepressants. By contrast, stress, depression, ageing and disease can have the opposite effect, triggering neurons to break down and even die. The ability of the brain to change in response to experience is known as structural plasticity, and it is in a tug-of-war with processes that drive neurodegeneration.

Structural plasticity occurs in other species too: for example, it was described in the fruit fly more than a quarter of a century ago. Yet, the molecular mechanisms underlying structural plasticity remain unclear. Li et al. now show that, in fruit flies, this plasticity involves Toll receptors, a family of proteins present in the brain but best known for their role in the immune system.

Fruit flies have nine different Toll receptors, the most abundant being Toll-2. When activated, these proteins can trigger a series of molecular events in a cell. Li et al. show that increasing the amount of Toll-2 in the fly brain makes the brain produce new neurons. Activating neurons in a brain region has the same effect, and this increase in neuron number also depends on Toll-2. By contrast, reducing the amount of Toll-2 causes neurons to lose their projections and connections, and to die, and impairs fly behaviour.

Li et al. also show that each Toll receptor has a unique distribution across the fly brain. Different types of experiences activate different brain regions, and therefore different Toll receptors. These go on to trigger a common molecular cascade, but they modulate it such as to result in distinct outcomes. By working together in different combinations, Toll receptors can promote either the death or survival of neurons, and they can also drive specific brain cells to remain dormant or to produce new neurons.

By revealing how experience changes the brain, Li et al. provide clues to the way neurons work and form; these findings may also help to find new treatments for disorders that change brain structure, such as certain psychiatric conditions. Toll-like receptors in humans could thus represent a promising new target for drug discovery.

Primary candidates to regulate brain plasticity are the neurotrophins. In the mammalian brain, neurotrophins (BDNF, NGF, NT3, NT4) regulate cell proliferation, cell survival, circuit connectivity, synaptic transmission and potentiation (*Lu et al., 2005*). Alterations in neurotrophins underlie brain disease, and anti-depressants increase the levels of the neurotrophin BDNF (*Krishnan and Nestler, 2008*; *Wohleb et al., 2016*). NTs have dual functions, as they promote plasticity via p75$^{NTR}$ activating NF-κB, and via Trk receptors activating AKT, ERK and CREB downstream, and they promote neurodegeneration via p75$^{NTR}$ and JNK signalling (*Lu et al., 2005*). *Drosophila* neurotrophins (DNTs) also regulate neuronal survival and death, connectivity and synaptic structural plasticity (*Zhu et al., 2008*; *Sutcliffe et al., 2013*; *McIlroy et al., 2013*; *Foldi et al., 2017*; *Ulian-Benitez et al., 2017*). However, there are no canonical tyrosine-kinase-Trk and p75$^{NTR}$ receptors in *Drosophila*, and instead, DNTs are ligands for the Kekkons, kinase-less members of the Trk family, and Tolls (*McIlroy et al., 2013*; *Foldi et al., 2017*; *Ulian-Benitez et al., 2017*). *Drosophila* Toll and mammalian Toll-Like-Receptors (TLRs) are best known for their universal function in innate immunity (*Leulier and Lemaitre, 2008*), but also have non-immune functions in development and in the central nervous system (CNS)(*Anthoney et al., 2018*). In neurons, Tolls and TLRs can promote neuronal survival via MyD88 and neuronal death via Sarm, both in flies and mammals (*Kim et al., 2007*; *McIlroy et al., 2013*; *Mukherjee et al., 2015*; *Foldi et al., 2017*). In humans, alterations in TLR function underlie brain diseases from stroke and neurodegeneration to multiple sclerosis and neuroinflammation (*Okun et al., 2011*; *Hanamsagar et al., 2012*). Most attention has focused on TLR functions in microglia, their response to damage or infection, and in neuroinflammation (*Fiebich et al., 2018*). However, TLRs are also in neurons, but functions in neurons and neural progenitor cells are largely unknown. Importantly, TLRs can influence neurogenesis, neuronal survival and death, neurite growth, synaptic transmission and behaviour, including learning and memory (*Ma et al., 2006*; *Rolls et al., 2007*; *Okun et al., 2010b*; *Okun et al., 2011*; *Qi et al., 2011*;

*Okun et al., 2012*; *Madar et al., 2015*; *Liu et al., 2016b*; *Patel et al., 2016*; *Hung et al., 2018*; *Min et al., 2018*). These findings suggest that TLRs could regulate structural brain plasticity, but this remains little explored.

Tolls regulate cell number plasticity in the *Drosophila* ventral nerve cord (VNC) through a three-tier mechanism (*Foldi et al., 2017*). In embryos and larvae, *Toll-6* and *Toll-7* maintain neuronal survival via MyD88 and NF-κB (*McIlroy et al., 2013*; *Foldi et al., 2017*). However, in pupae, they can also promote apoptosis via Weckle (Wek), Sarm and JNK (*Foldi et al., 2017*). Furthermore, different Tolls lead to different outcomes, for instance, Toll-1 is more pro-apoptotic than Toll-6 (*Foldi et al., 2017*). Whether a neuron lives or dies in the CNS depends on the ligand and its cleavage state it receives, the *Toll* or combination of *Tolls* it expresses, and the downstream adaptors available for signalling (*Foldi et al., 2017*). Thus, cell number control is context dependent. The ability of DNTs and Tolls to regulate cell number by promoting both cell survival and cell death is crucial for the modulation of structural brain plasticity, homeostasis and neurodegeneration.

Here, we asked whether Toll receptors influence developmental and structural plasticity in the *Drosophila* brain.

## Results

### A Toll receptor map in the *Drosophila* brain

To find out whether *Toll* receptors are expressed in the brain, we looked for *Toll* transcripts in embryos and dissected CNSs from larvae to adult brains, using reverse-transcription PCR (RT-PCR) (*Figure 1—figure supplement 1*). *Toll-3* transcripts were absent from larval L2 CNSs; *Toll-4* and −9 mRNAs were barely detected in all sample types; whereas *Toll-1,–2, −5,–6, −7,–8* were expressed in embryos, larval (L2, L3) CNSs, and pupal and adult fly heads (*Figure 1—figure supplement 1*). Thus, all *Tolls* are expressed in pupal and adult brains, with *Toll-1,–2, −5,–6, −7,–8* most prominently.

To visualise the spatial distribution of Tolls in the brain, we generated *GAL4* reporter lines for the *Tolls*. Using CRISPR/Cas9-accelerated homologous recombination to insert a *pTV* cassette (*Baena-Lopez et al., 2013*), we generated a knock-in/knock-out *Toll-2$^{pTV}$* allele, and we used the *pTV-attP* landing site to generate a *Toll-2$^{pTV}$-GAL4* driver line. *Toll-4GAL4* and *Toll-5GAL4* were generated by CRISPR/Cas9, inserting *T2AGAL4* immediately upstream of the start codon. Unfortunately, we could not get transformants for *Toll-9*. *Toll-3GAL4*, *Toll-6GAL4* and *Toll-7GAL4* were made using Recombinase-Mediated Cassette Exchange (RMCE) of *MIMIC* insertions into the intronless coding regions of these genes. *Toll-8GAL4* is *Tollo$^{MD806}$*, which has a P-element insertion just 180 bp upstream of the start codon, within the 5'UTR of *Toll-8*. The GAL4 driver lines were used to visualise membrane tethered FlyBow (for *Toll-2,–4, −5,–7*) and tdTomato (for *Toll-3,–6, −8*) reporters, and all necessarily reproduced the endogenous expression patterns of the *Toll* genes. Toll-1 was visualised using commercially available and previously validated anti-Toll-1 antibodies (*Lund et al., 2010*; *Khadilkar et al., 2017*).

In the adult brain, Toll-1 was found in all photoreceptor cells (*Figure 1A,G*). *Toll-2,–5, −6,–7* and −8 were all expressed in Kenyon cells, with *Toll-2* and −6 comprising most cells (*Figure 1A,B,C*). *Toll-5* and −7 were expressed in the protocerebral bridge (*Figure 1A,B*). *Toll-2,–5, −6,–7* and −8 were differentially expressed in the antennal lobes (*Figure 1D*). *Toll-1,–2, −3,–5, −6,–7, −8* were expressed in distinct and overlapping fan shaped body neuropile layers (*Figure 1E*); *Toll-1,–2* and −7 in distinct ellipsoid body neuropile rings (*Figure 1F*), and *Tol-1,–2, −6* and −8 in the sub-esophageal ganglion (SOG) (*Figure 1A*). *Toll-2,–3, −5,–6, −7,–8* were expressed in optic lobes, with *Toll-3* and −6 having a prominent expression in the lamina (*Figure 1A,G*) and *Toll-2* a broad expression throughout the optic lobes (*Figure 1A,G*). In summary, these patterns revealed: (1) a map of *Toll* expression profiles coincident with anatomical brain domains; (2) profiles specific to each *Toll*; (3) complementary patterns in neuropiles of the visual system and central complex (fan shaped body, ellipsoid body and protocerebral bridge); (4) overlapping distributions in optic lobes, antennal lobes, Kenyon cells and mushroom bodies. Tolls could influence brain structure and connectivity by virtue of their topographic profiles (*Figure 1I*).

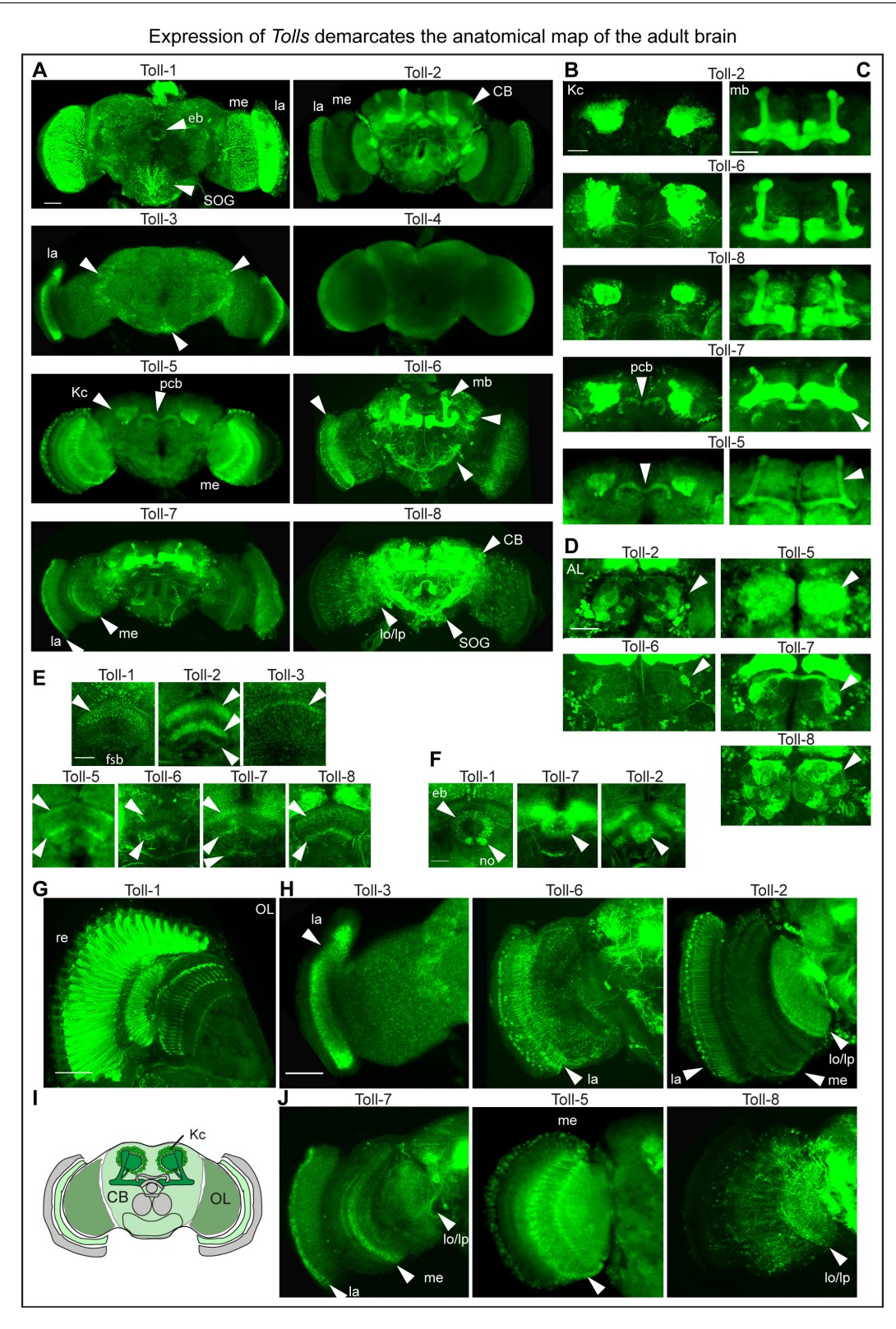

**Figure 1.** Expression of *Tolls* demarcates the anatomical map of the adult brain. (**A**) *Toll* receptor expression visualised with: Anti-Toll-1 antibodies, in retinal photoreceptors; CRISPR/Cas9 generated *Toll-2*^pTV*GAL4 > FlyBow*, throughout the brain; MIMIC-RMCE generated *Toll-3GAL4 > tdTomato* restricted to the lamina; CRISPR/Cas9 generated *Toll-4GAL4 > FlyBow* did not reveal any signal; CRISPR/Cas9 generated *Toll-5GAL4 > FlyBow* was prominent in medulla, Kenyon cells and protocerebral bridge (pb). (Unfortunately, we could not get CRISPR/Cas9 data for *Toll-9*). MIMIC-RMCE

*Figure 1 continued on next page*

*Figure 1 continued*

generated *Toll-6GAL4 > tdTomato* is prominent in lamina, Kenyon cells and central brain interneurons. MIMIC-RMCE generated *Toll-7GAL4 > FlyBow* was prominent in optic lobe and Kenyon cells. *Toll-8GAL4^{MD806} > tdTomato* was prominent in central brain and Kenyon cells. (**B–G**) Higher magnification views to show signal in: (**B**) Kenyon cells (KCs); (**C**) mushroom bodies (mb); (**D**) antennal lobes (AL); (**E**) fan shaped body (fsb); (**F**) ellipsoid body (eb); (**G,H,J**) optic lobes (OL). (**I**) Drawing illustrating the brain domains of KCs, central brain (CB) and OLs used for the functional analysis. La: lamina; me: medulla; lo: lobula; lp: lobula plate; no: noduli; pcb: protocerebral bridge. Scale bars: A,E,F,G: 25 µm; B,C,D,H,J: 50 µm. For genotypes and sample sizes see Materials and methods and *Supplementary file 2*. See *Figure 1—figure supplement 1*.

The online version of this article includes the following figure supplement(s) for figure 1:

**Figure supplement 1.** Most Tolls are expressed in the central nervous system.

## Toll-2 is neuro-protective in the brain

To ask whether Tolls may influence brain development and/or adult structural brain plasticity, we focused on *Toll-2*, as it is most broadly expressed. In neurons, *Drosophila* Tolls and mammalian TLRs promote neuronal survival via MyD88 and neuronal death via Sarm (*Kim et al., 2007*; *McIlroy et al., 2013*; *Mukherjee et al., 2015*; *Foldi et al., 2017*; *Figure 2C*). Thus, to investigate whether Toll-2 is required for cell survival in brain development, we first verified whether MyD88 was expressed in the brain. *MyD88^{NP6394}* flies bear a *GAL4* insertion within the transcribed 5'UTR exon, and thus it necessarily represents the endogenous expression pattern of the gene (from now on called *MyD88GAL4*). *MyD88GAL4 >tdTomato* revealed MyD88+ cells throughout the optic lobes, central brain, and mushroom bodies (*Figure 2A*). *Toll-2* appears to be expressed in all Kenyon cells, whereas *MyD88* is only in the subset that projects along the core α,β lobes (*Figure 2A*). Using the nuclear reporter histone-YFP (his-YFP) revealed that more cells expressed *Toll-2* than *MyD88* (*Figure 2B*). In the optic lobes, *MyD88 >hisYFP* includes large, sparsely distributed cells, that may or may not also be Toll-2+ (*Figure 2B*). To identify the Toll-2+ and MyD88+ cells, *Toll-2 >hisYFP* and *MyD88 >hisYFP* adult brains were labelled with pan-neuronal anti-Elav and pan-glial anti-Repo. There were many MyD88+ Elav+ neurons as well as MyD88+ Repo+ glia (*Figure 2D*). By contrast, none of the Toll-2+ cells were Repo+, whilst most Toll-2+ cells were Elav+ (*Figure 2E*). Thus, MyD88+ cells comprise both neurons and glia, that are most likely regulated by multiple Tolls, and Toll-2+ cells are mostly neurons.

To ask whether Toll-2 might regulate cell survival in brain development, we visualised apoptotic cells using anti-Dcp1 antibodies upon *Toll-2* knock-down. In brain development, the peak of cell death occurs 24 hr after puparium formation (*Hara et al., 2018*). So, we quantified apoptosis in day one pupal brains, using purposely developed DeadEasy Central Brain software. *Toll-2* RNAi knock-down in MyD88+ cells increased apoptosis in the pupal central brain (*Figure 2G*), meaning that Toll-2 is required to maintain cell survival. To verify whether apoptosis resulted in cell loss, we counted automatically *MyD88 >hisYFP* cells in the central brain. Using two independent *UAS-Toll-2* RNAi lines of flies, *Toll-2* knock-down with *MyD88GAL4* decreased cell number in the central brain of both pupae and adult flies (*Figure 2H*). Thus, *Toll-2* loss of function in MyD88+ cells increased apoptosis and caused cell loss. On the other hand, sustained over-expression of *Toll-2* with *MyD88GAL4* throughout development did not affect cell number in the pupal or adult brains (*Figure 2H*). In larvae and pupae, Tolls can also induce apoptosis via Sarm, and different Tolls have distinct pro-apoptotic drive (*Foldi et al., 2017*). As *Toll-2* gain of function did not reduce cell number, this meant that Toll-2 does not induce apoptosis in the pupal or adult brain. Together, these data showed that *Toll-2* maintains the survival of MyD88+ neurons during brain development.

To test if *Toll-2* maintains neuronal survival via the MyD88 pathway, we knocked-down *MyD88* with *MyD88GAL4*. Similarly to *Toll-2* loss of function, *MyD88* knock-down also resulted in cell loss in the pupal central brain (*Figure 2H*). However, by the adult stage, cell number was restored vs. controls (*Figure 2H*). This was in contrast to the persistent cell loss caused by *Toll-2-RNAi* into the adult, suggesting that MyD88 carries out further functions too. These data showed that MyD88 is required to maintain cell survival during brain development, downstream of at least Toll-2.

To analyse the effect in Kenyon Cells (KCs), we developed another plug-in - DeadEasy KCs- to count KCs labelled with *Toll-2 >hisYFP*. *Toll-2^{pTV}/Toll-2^{Δ7-35}* mutations increased KC number in pupal brains, but neither sustained gain nor loss of *Toll-2* function with *Toll-2 >Toll-2RNAi* knock-down or *Toll-2^{pTV}/Toll-2^{Δ7-35}* mutations affected KC number in adult brains (*Figure 2I*). Over-expression of

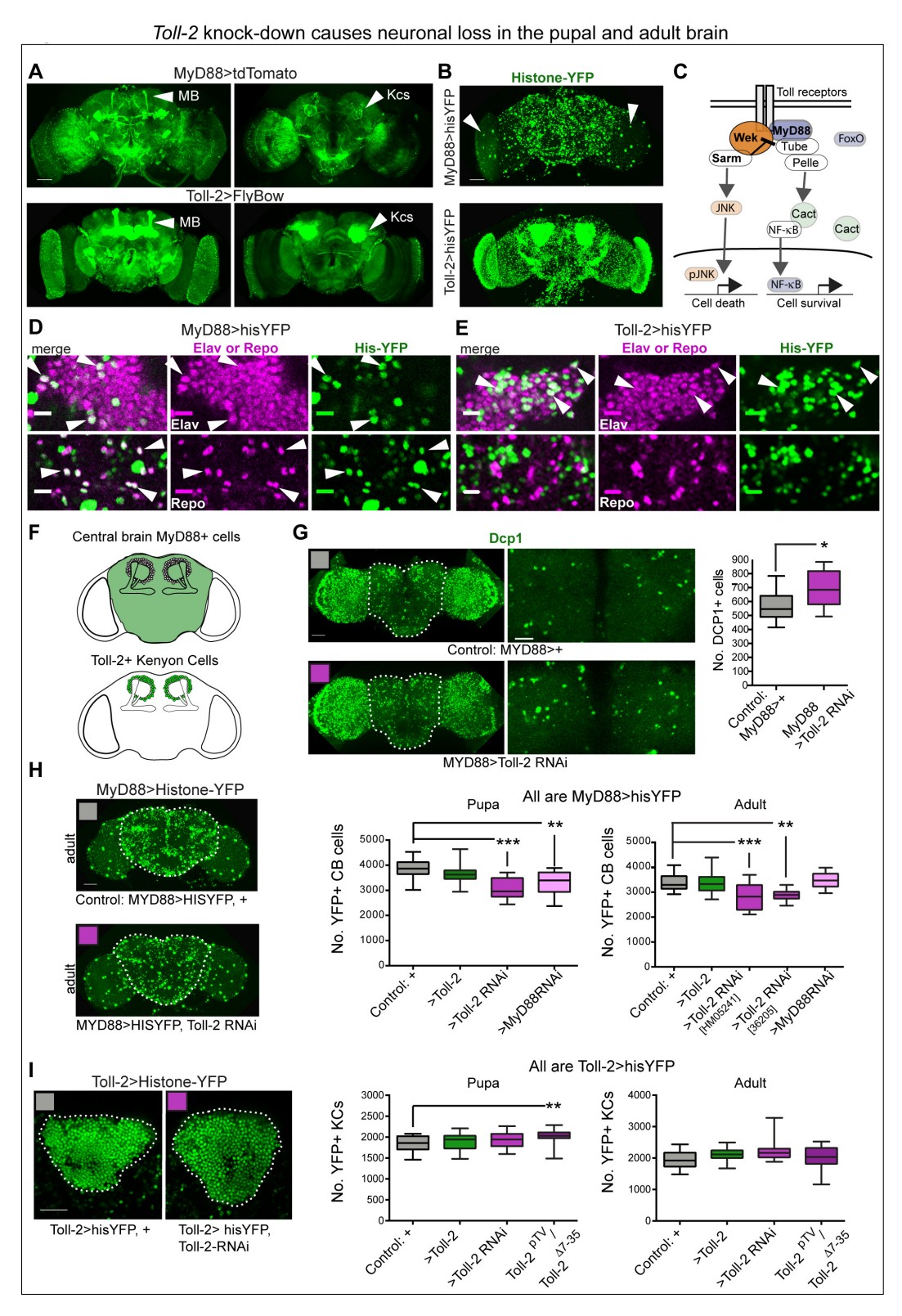

**Figure 2.** *Toll-2* knock-down caused neuronal loss in the pupal and adult brain. (**A,B**) Overlapping but distinct expression of *Toll-2* and the adaptor *MyD88*, visualised with *MyD88 >tdTomato, Toll-2^{pTV}GAL4 > UASFlyBow, MyD88 >histone* YFP and *Toll-2^{pTV}GAL4 > UAShistone YFP. Toll-2* is expressed in more Kenyon cells (arrowheads) than *MyD88*. Note the large MyD88+ cells in the optic lobes (B, arrowhead). (**C**) Diagram of signalling pathways downstream of Toll receptors regulating cell death and cell survival (adapted from ***Foldi et al., 2017***). (**D,E**) The pan-neuronal marker anti-
*Figure 2 continued on next page*

*Figure 2 continued*

Elav co-localises with His-YFP in both MyD88+ and Toll-2+ cells, whereas the pan-glial marker anti-Repo only co-localises with MyD88 >His-YFP+ cells (arrowheads). (F) Drawings showing in green the central brain region of interest (ROI), and Kenyon cells (KCs), used for automatic cell counting with DeadEasy. (G) *Toll-2 RNAi* knock-down increased the number of anti-Dcp1+ apoptotic cells, in day one pupal central brains (dashed line indicates ROI); cells quantified automatically in the ROI in 3D throughout the stack of images, with DeadEasy software. Left: full projection; right: projection of five optical sections only (5 µm). Quantification in box-plot graph: Student t-test p=0.0295. (H) *Toll-2 RNAi* knock-down decreased MyD88+ cell number in pupal and adult brains, latter using two independent RNAi lines; *MyD88* RNAi knock-down also decreased cell number in the pupal brain (left), but not in the adult brain (right). Dashed lines in (H) indicate central brain ROI used for automatic counting of MyD88 >hisYFP+ cells with DeadEasy Central Brain software. Box-plots: Left: One Way ANOVA p<0.001, and right p<0.0001, post-hoc Dunnett tests. (I) Neither *Toll-2* over-expression nor RNAi knock-down altered KC number, in pupal nor adult brains, but *Toll-2$^{pTV}$/Tollp2$^{\Delta7-35}$* mutants had more KCs. Dashed lines in (I) indicate ROI counted automatically with DeadEasy Kenyon Cells software, box-plot graphs on right: Kruskal Wallis ANOVA, both p>0.1. Scale bars: A,B,G left,H: 50 µm; D,E: 10 µm; G right, I: 25 µm. For genotypes, sample sizes and statistical details, see *Supplementary file 2*. *p<0.05, **p<0.01, ***p<0.001. See *Figure 2— figure supplement 1*.

The online version of this article includes the following figure supplement(s) for figure 2:

**Figure supplement 1.** Over-expression of *Toll-2* does not affect Kenyon cell number.

*Toll-2* with another mushroom body driver, *MBGAL4,* did not affect KC number in pupal or adult brains either (*Figure 2—figure supplement 1B,C*). Thus, KCs are resilient to alterations in *Toll-2* function alone.

To further test whether Toll-2 is neuroprotective, we induced *Toll-2$^{pTV}$* homozygous mutant MARCM clones. Genetic complementation tests over the previously described null allele *18w$^{\Delta7-35}$* (*18* w is a synonym of *Toll-2*, thus hereby will be referred to as *Toll-2$^{\Delta7-35}$*) and a deficiency for the locus, *Df(2R)BSC594*, showed that *Toll-2$^{pTV}$* is a strong hypomorphic loss of function allele (*Figure 3—figure supplement 1A*). *Toll-2$^{pTV}$* mutant clones were induced from dividing cells in the pupal brain, where *Toll-2* is widely expressed (*Figure 3A*). They were induced using *hsFlp*, and resulting mutant neurons were visualised in adult brains with *elav >mCD8* GFP (*Figure 3B*). Loss of *Toll-2* function caused extensive neuronal loss, neuronal atrophy, loss of neurites - axons and dendrites- and axonal misrouting (*Figure 3C–H*). Loss of dendrites could be clearly observed in the lamina (*Figure 3C*); axonal degeneration and misrouting in medulla and lobula (*Figure 3D*); and loss of entire axonal neuropiles in the medulla, SOG and fan shaped body (*Figure 3D,E,H*). Whether mushroom bodies were affected was less clear (*Figure 3F,G*), perhaps because our heat shock regime missed mushroom body neuroblast divisions, as we could not observe many mushroom body projections in control brains either (*Figure 3G*). Dramatic neuronal deficits could be found throughout many brain domains (*Figure 3C–H*). The loss of neurons in mutant clones was consistent with Toll-2 maintaining neuronal survival, but could also reflect a function promoting progenitor cell proliferation; neurite atrophy meant that *Toll-2* loss of function prevented neuronal differentiation or caused neurodegeneration.

*Toll-2* mutants are semi-lethal, have reduced lifespan and impaired climbing (*Figure 3—figure supplement 1B,C*) – phenotypes commonly associated with neurodegeneration. *Toll-2* is expressed in the visual system, ventral nerve cord and central complex, which is the higher control center for locomotion and spatial navigation in the brain (*Strauss and Heisenberg, 1993*). Thus, we tested the performance of *Toll-2* mutants in the Buridan arena, which could reveal whether loss of *Toll-2* affected vertical vs. horizontal locomotion, visual processing, or motivation to walk. Wild-type flies free to walk in a circular lit-up arena walked back and forth between two diametrically opposed dark stripes (38.5%), but most often wondered randomly (61.5%, *Figure 3I*). *Toll-2$^{pTV}$/Df(2R)BSC22* and *Toll-2$^{\Delta7-35}$/Df(2R)BSC22* mutants also most often walked randomly or along the perimeter (63.7–92%), but overall walked less than controls, and some walked very little (7.7–18.1%). Adult specific *Toll-2* knock-down in neurons, with *tubulinGAL80$^{ts}$* to switch on GAL4 and drive *elav >Toll-2RNAi* after adult fly eclosion, reproduced the behavioural phenotypes of the mutants (*Figure 3I*). Importantly, this shows that *Toll-2* is required in adult neurons. Both wild-type and mutant flies could walk between the black stripes, meaning that loss of *Toll-2* function does not impair vision. Quantitative analysis of the flies' walking behaviour did not reveal significant differences between the genotypes in their preference to walk between the dark stripes, or away from the centre of the arena. Hence, we cannot conclude that *Toll-2* mutant flies have impaired visual processing. Interestingly, all wild-type flies walked more than *Toll-2* mutants. In fact, loss of *Toll-2* function significantly affected

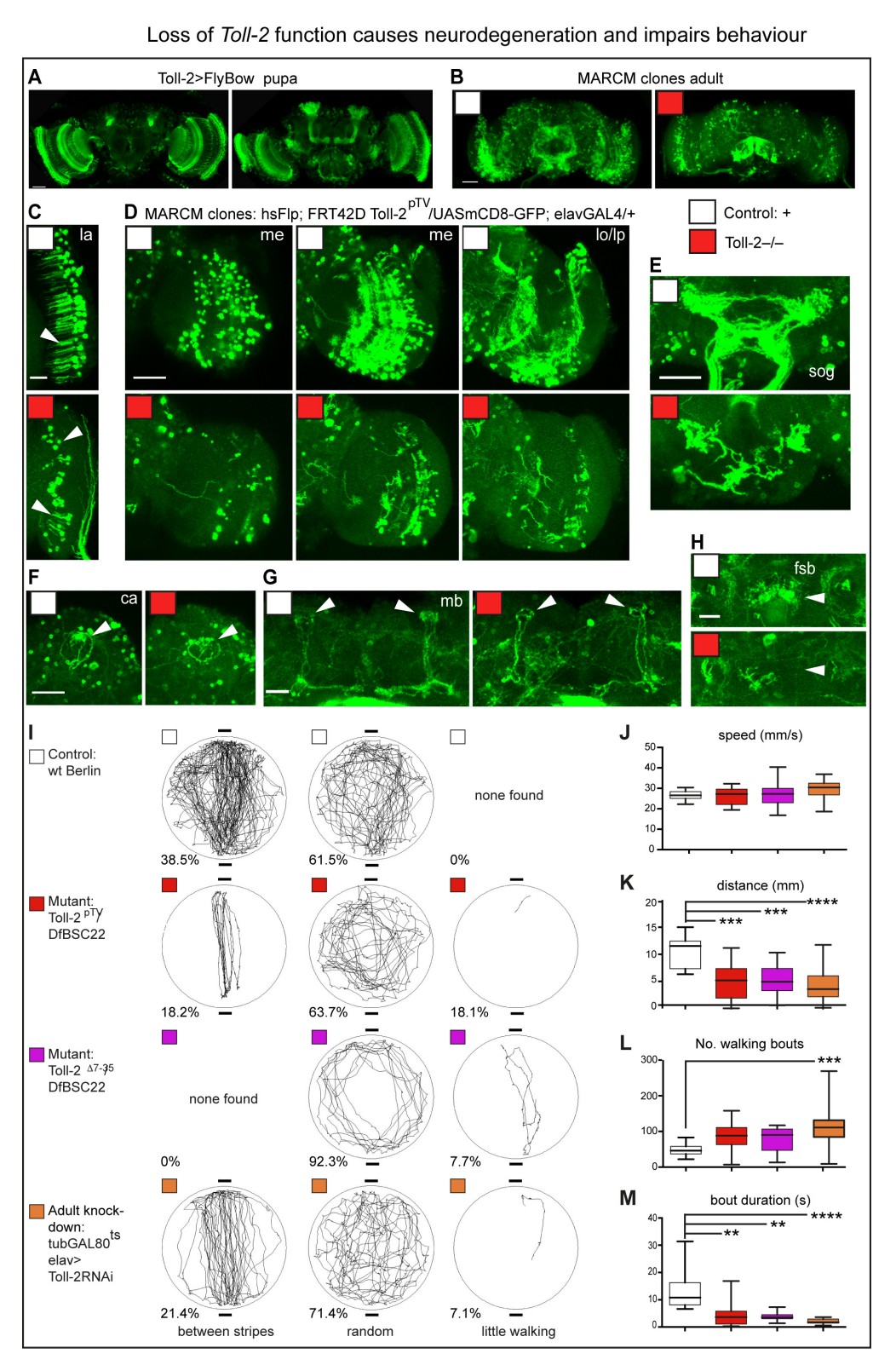

**Figure 3.** Loss of *Toll-2* function caused neurodegeneration and impaired behaviour. (A) *Toll-2* is expressed in pupal brains, most prominently in optic lobes, mushroom bodies and SOG. (**B–H**) *Toll-2^pTV* mutant MARCM clones - labelled with GFP - induced in pupa cause neuronal loss throughout the adult brain (genotype of clones: *elavGAL4, UASmCD8GFP, hsFlp; neo FRT42D Toll-2^pTV/neo FRT42D Toll-2^pTV*). Compare controls with *Toll-2* mutant clones in: (**B**) whole brains; (**C**) loss of neurons and dendrites (arrow) in lamina (la); (**D**) neuronal loss, misrouted and/or aberrant axons in optic lobe; (**E**)
*Figure 3 continued on next page*

*Figure 3 continued*

loss of sub-esophageal ganglion (SOG) neuropile; (**F,G**) Calyx (ca) and mushroom bodies (mb) were less affected; (**H**) loss of fan shaped body (fsb) neuropiles. me: medulla; lo: lobula; lp: lobula plate. (**I–M**) Buridan arena behavioural assay revealed impaired locomotion with *Toll-2^pTV* loss of function. (**I**) Representative trajectories of single flies filmed for the same amount of time, walking in a lit-up arena, with two diametrically opposed dark stripes. Phenotypes were divided into three categories – between stripes, random, little walking - and percentages indicate how many flies per genotype showed each phenotype (penetrance); no statistically significant differences were found. (**J–M**) Automatic measurement of locomotion parameters with purposely written software. *Toll-2* mutant flies walked less than controls: they achieved equal speeds as controls, but they had more walking bouts that were brief, and thus walked shorter distances. Box-plots, quantifications: (**J, K, L**) One Way ANOVA: (**K**) p<0.0001; (**L**) p<0.01; (**M**) Kruskal-Wallis ANOVA p<0.0001; stars indicate post-hoc (**J,K,L**) Dunnett and (**M**) Dunn's test comparisons to fixed controls. Scale bars: A,B,D,E,F: 50 µm; C,G,H: 25 µm. *p<0.05; **p<0.01; ***p<0.001; ****p<0.0001. For genotypes, sample sizes and statistical details, see *Supplementary file 2*. See *Figure 3—figure supplement 1*.

The online version of this article includes the following figure supplement(s) for figure 3:

**Figure supplement 1.** *Toll-2* loss of function compromises survival, longevity and locomotion.

locomotion: all genotypes could walk at the same speed (*Figure 3J*), but *Toll-2* mutants spent less time walking than controls, thus overall travelled shorter distances (*Figure 3K*), and although they had as many or more walking bouts, these were brief (*Figure 3L,M*). Importantly, these phenotypes were consistent across different *Toll-2* mutant alleles, and also when *Toll-2* was conditionally knocked-out in adult post-mitotic neurons only (*Figure 3J–M*). As *Toll-2* mutants could achieve the same speeds as wild-type flies, but walked less, either motor circuit function and/or the motivation to walk were impaired.

To conclude, *Toll-2* loss of function resulted in neurodegeneration and impaired behaviour. Thus, Toll-2 is required for the formation and integrity of brain neural circuits.

## Toll-2 can increase cell number in the adult brain

To ask whether *Toll-2* might affect structural plasticity in the adult brain, we altered its function at the adult critical period (i.e. day 0–5 post-eclosion), when the brain is most plastic (*Technau, 1984*; *Heisenberg et al., 1995*; *Barth and Heisenberg, 1997*; *Barth et al., 1997*). We used *tubulin-GAL80^ts* to silence *GAL4*, then switched on *GAL4* at post-eclosion adult day 0 and analyzed the brains two days later. Over-expression of *Toll-2* with *MyD88GAL4* increased the number of histone-YFP+ cells in the central brain (*Figure 4A*), meaning that Toll-2 can regulate cell number at the adult critical period. Conditional *Toll-2*-RNAi-knock-down also caused a mild increase in cell number (*Figure 4A*), which could be due to compensatory adjustments by other Tolls. In mushroom body KCs, neither *Toll-2* knock-down nor over-expression with *Toll-2^ptv GAL4* restricted to the critical period had any effect on KC number (*Figure 4B*). The optic lobes are particularly susceptible to structural plasticity (*Heisenberg et al., 1995*; *Barth et al., 1997*), thus we drove conditional over-expression with *tubulinGAL80^ts* and *Toll-2 >histone* YFP, and automatically counted YFP+ cells with purposely adapted DeadEasy Optic Lobes software (*Figure 4C*). Conditional *Toll-2-RNAi* knock-down had no effect, whereas over-expression of *Toll-2* increased the number of YFP+ medulla neurons (*Figure 4C*). Thus, Toll-2 can increase cell number in the adult optic lobes. The increase in cell number by *Toll-2* gain of function in the central brain and optic lobes is consistent with a neuroprotective function, but could also involve cell proliferation. Either way, these data showed that Toll-2 is not pro-apoptotic in the adult brain, and instead can positively regulate cell number during the adult critical period.

Experience increased the volume of multiple domains of the adult brain (*Technau, 1984*; *Heisenberg et al., 1995*; *Barth and Heisenberg, 1997*; *Barth et al., 1997*). To test whether stimulating neuronal activity during the critical period affects cell number in the adult brain, we activated neurons using the heat sensitive TrpA1 cation channel, whilst preventing leakage and excito-toxicity, and automatically counted His-YFP+ cells (*Figure 4D*). Neuronal activation with TrpA1 increased *Toll-2 >hisYFP* medulla neuron number (*Figure 4D*). Importantly, this increase could be rescued with *Toll-2-RNAi* knock-down (*Figure 4D*). This meant that neuronal activity alters cell number in the optic lobes, via a Toll-2 dependent mechanism.

To conclude, both Toll-2 and neuronal activity can increase cell number in the adult brain. KC number is robustly unaltered, but cell number in the central brain and optic lobes is plastic, and plasticity depends on Toll-2.

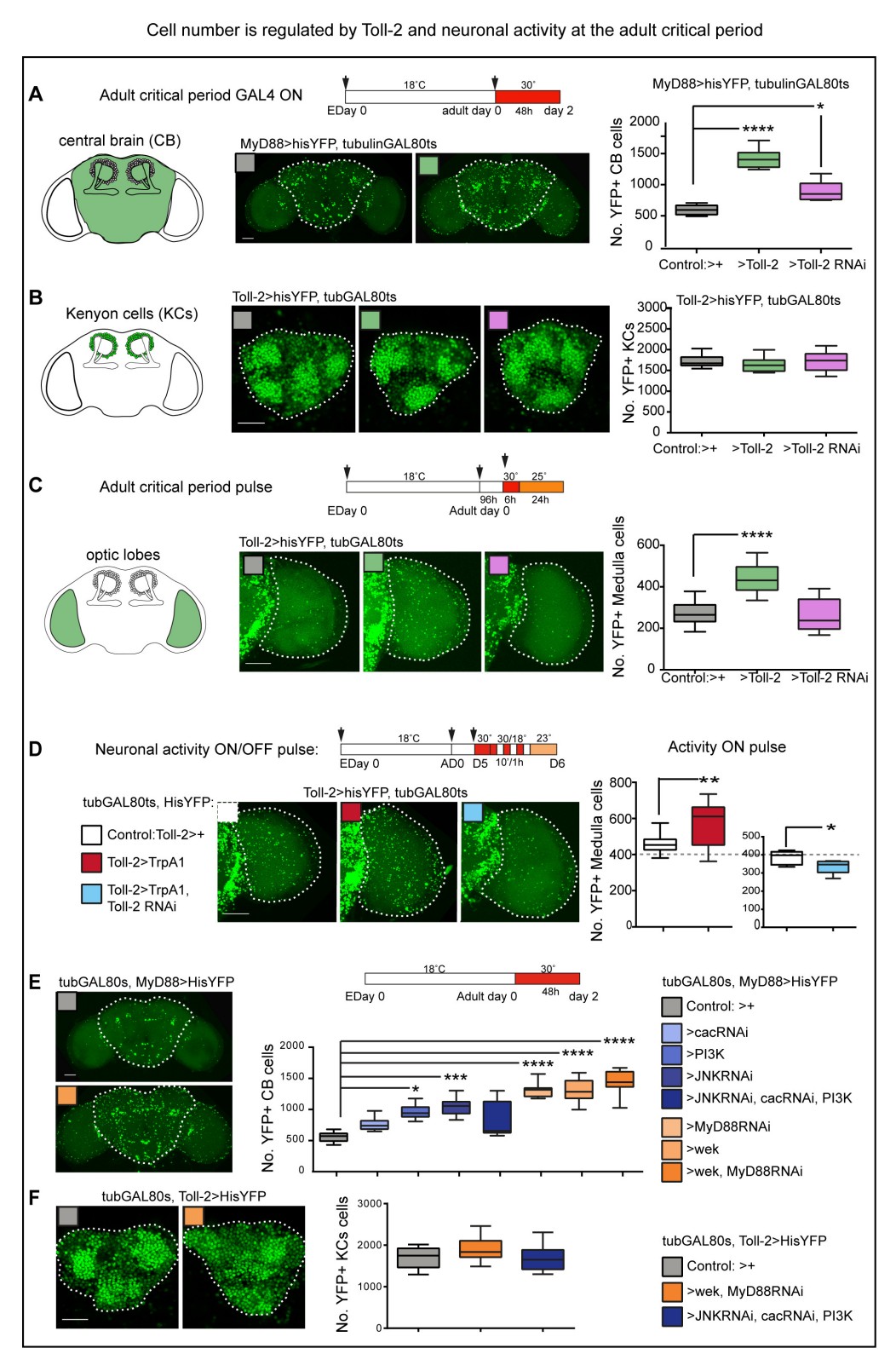

**Figure 4.** Cell number is regulated by Toll-2 and neuronal activity at the adult critical period. At adult days 0–2 post-eclosion, within the critical period: (A) Conditional over-expression of *Toll-2* increased MyD88 >hisYFP+ cell number in the central brain. Cells were counted automatically in 3D throughout the whole stack with DeadEasy Central Brain, dashed line in all figures indicates ROI quantified. Box-plots, Kruskal-Wallis p<0.0001, post-hoc Dunn test; (B) neither conditional over-expression nor knock-down of *Toll-2*, altered Toll-2 >hisYFP+ Kenyon cell number, counted automatically
*Figure 4 continued on next page*

*Figure 4 continued*

with DeadEasy Kenyon Cells, box-plots; (C) conditional over-expression of *Toll-2* increased Toll-2 >hisYFP+ cell number in the optic lobe medulla, counted automatically with DeadEasy Optic Lobe. Box-plots, One Way ANOVA p<0.0001, post-hoc Dunnett; (D) pulses of neuronal activation with TrpA1 increased Toll-2 >hisYFP+ cell number in the medulla, and this could be rescued with *Toll-2* RNAi knock-down. Box-plots: Left: Un-paired Student t-test, p=0.0058; Right: Un-paired Student t-test, p=0.0225. (E) Knocking-down *JNK* and *cactus* and over-expressing *activated PI3K* (*UAS-Dp110CAAX*), alone or in combination, in MyD88+ cells with *tubGAL80^{ts}, MyD88GAL4* increased cell number in the central brain, consistent with pro-survival signalling downstream of Toll-2. However, over-expressing either *wek* or *MyD88* RNAi knock-down increased cell number in the central brain, and even further in combination, suggesting that Wek also has non-apoptotic functions that antagonise MyD88. Box-plots, Kruskal-Wallis ANOVA p<0.0001, post-hoc Dunn test. (F) By contrast, no statistically significant changes were detected in KCs upon manipulation of any of these downstream effectors, although a mild increase in cell number was observed with *UAS-wek, UASMyD88RNAi*. Box-plots, One Way ANOVA p=0.0354, post-hoc Dunnett. Dashed lines indicate regions of interest (ROI) for automatic cell counting with DeadEasy. Scale bars: A,C,D,E:50 μm; B,F:25 μm. For genotypes, sample sizes and statistical details, see *Supplementary file 2*. *p<0.05; **p<0.01; ***p<0.001; ****p<0.0001. See *Figure 4—figure supplement 1*.

The online version of this article includes the following figure supplement(s) for figure 4:

**Figure supplement 1.** Toll-signalling downstream factors were found in the brain.

## Known effectors of Toll signalling are distributed in developing and adult brains

The alterations in cell number caused by loss and gain of *Toll-2* function strongly implied that downstream nuclear signalling pathways were most likely involved. To test this, we visualised the distribution of the different adaptors and downstream targets of Toll signalling. Toll-receptor signalling positively regulates cell survival via Wek, MyD88, NF-κB, and cell death via Wek, Sarm, JNK (*Kim et al., 2007*; *McIlroy et al., 2013*; *Mukherjee et al., 2015*; *Foldi et al., 2017*); Toll-6 also functions upstream of pro-survival ERK, and FoxO, with nuclear FoxO promoting apoptosis whereas cytoplasmic export of FoxO by PI3Kinase signalling promotes cell survival (*Siegrist et al., 2010*; *McLaughlin et al., 2016*; *Foldi et al., 2017*). Sarm inhibits MyD88, and drives the pro-apoptotic functions of Tolls (*Carty et al., 2006*; *Mukherjee et al., 2015*; *Foldi et al., 2017*). Thus, we used: 1) available GAL lines *sarm^{NP7460}* and *sarm^{NP0257}*, to drive expression of the *FlyBow* reporter; (2) the JNK signalling reporter TRE-Red; (3) anti-FoxO antibodies: (4) GFP-tagged forms of the pro-survival NF-κB transcription factors Dorsal (Dl) and Dif (*Anthoney et al., 2018*), which function downstream of MyD88. Both *dorsal* and *dif* produce cytoplasmic isoforms that lack the nuclear localization signal, and nuclear isoforms that have it (*Zhou et al., 2015*). We used transgenic flies bearing Bacmids in which the nuclear isoforms were tagged with GFP, Dif-GFP-FPTB and Dl-GFP-FPTB. All of these signalling reporters were found both in pupal and adult brains (*Figure 4—figure supplement 1*). Thus, Toll signalling adaptors MyD88 and Sarm, and their downstream targets that normally regulate gene expression, cell survival and cell death, could function in pupal and adult brains.

## Toll-2 can promote cell proliferation in the pupal and adult brain

So far, data were consistent with Toll-2 maintaining neuronal survival via the canonical MyD88 pathway in the brain. However, multiple Tolls can regulate this pathway, and Tolls can also promote apoptosis via non-canonical signalling pathways (*Foldi et al., 2017*), thus altering the levels of Toll-2 could cause compensation by other Tolls, compounding the phenotypes. Thus, we tested how signalling downstream of Tolls would affect cell number in the adult brain.

To activate the pro-survival pathways downstream of Tolls in either MyD88+ central brain cells or Toll-2+ KCs, we knocked-down pro-apoptotic JNK signalling, activated pro-survival NF-κB signalling by knocking-down the inhibitor *cactus*, and induced the nuclear export of FoxO by over-expressing the activated form of PI3Kinase (*McLaughlin et al., 2016*; *Foldi et al., 2017*), and used *tubulin-GAL80^{ts}* to induce *GAL4* expression conditionally, in adults only (*Figure 4E*). Pro-survival signalling at the adult critical period increased cell number in the central brain (*Figure 4E*), consistent with the neuroprotective function of Toll-2. To drive Toll-dependent pro-apoptotic signalling at the adult critical period, we knocked-down *MyD88* and over-expressed *wek*, which links Tolls to pro-apoptotic Sarm and JNK signalling (*Foldi et al., 2017*). This genotype results in dramatic cell loss in pupae (*Foldi et al., 2017*). Unexpectedly, over-expression of *wek*, or *MyD88* RNAi knock-down, or both together, did not cause cell loss in adult day two central brains. Instead, they all increased cell number, with the combination of *wek* gain of function and *MyD88* loss of function having the greatest

effect (*Figure 4E*). This surprising result indicated that: (1) *MyD88* loss of function enabled an alternative pathway to increase cell number; (2) Wek did not simply induce apoptosis in the adult brain; (3) removal of *MyD88* facilitated a hitherto unknown Wek function. Perhaps Wek-induced apoptosis provoked non-autonomous compensatory proliferation of some cells, or Wek itself induced cell proliferation, and this was antagonized by MyD88.

In KCs, pro-survival signalling had no effect on Toll-2+ cells (*Figure 4F*), and over-expression of *wek* with *MyD88* knock-down caused a rather mild, not significant increase in cell number (*Figure 4F*). This mild increase indicated that different subsets of KCs could be regulated via different mechanisms and/or redundant functions between multiple Tolls might enable compensatory adjustments.

The above surprising results raised an important question: did Toll-2 gain of function increase cell proliferation in the adult brain? To test whether Toll-2 might induce proliferation in the adult brain, we asked whether over-expression of the cell cycle inhibitor, *Retinoblastoma-protein factor (Rbf²⁸⁰)*, could influence Toll-2+ cell number. We over-expressed *Rbf²⁸⁰* with *Toll-2GAL4* and this caused lethality at day one pupa, precluding analysis of adult brains. In pupae, over-expression of *Rbf²⁸⁰* decreased Toll-2+ KC number (*Toll-2 >hisYFP, Rbf²⁸⁰*, *Figure 5A*), showing that Toll-2 functions in mushroom body neuroblasts, that divide to produce KCs. To test whether Toll-2 might influence proliferation in other brain domains, we asked whether blocking proliferation with *Rbf²⁸⁰* could rescue the increase in *his-YFP+* cell number caused by *Toll-2* gain of function. And it could, *Rbf²⁸⁰* rescued the excess in cell number caused by *Toll-2* over-expression, both in central brain (with *MyD88 >hisYFP*) and optic lobes (with *Toll-2 >hisYFP*, *Figure 5A*). These data meant that Toll-2 can induce cell proliferation in pupal and adult brains.

To further test whether Toll-2 could induce cell proliferation in the adult brain, we asked whether over-expression of *Toll-2* restricted to the adult critical period could drive generation of mitotic recombination MARCM clones. Mitotic recombination was induced with heat-shock-Flipase at adult day one and neurons were visualised at adult day two with *elavGAL4 >UASmCD8* GFP; all neurons also over-expressed *Toll-2*, but control samples did not. We found no clones in control brains (n = 17), but amongst *Toll-2* over-expressing brains, 4/17 had GFP+ clones (*Figure 5B,C*). Importantly, the clones resulted in differentiated neurons that sent projections to different medulla layers, SOG, central brain and mushroom bodies (*Figure 5BC*). These data show that gain of *Toll-2* function induced cell proliferation in adult brains.

To visualise proliferating cells, we used a common readout of cell division - the G1 to S transition - with the S-phase marker PCNA-GFP. At 33 hr after puparium formation, there were some PCNA-GFP+ cells in normal control brains (*Figure 5D*), as previously reported (*Siegrist et al., 2010*). Heat-shock induced conditional over-expression of *Toll-2* in the pupa (*PCNAGFP, hsGAL4 >Toll-2*) increased the number of PCNA-GFP+ cells, in the central brain and mushroom bodies (*Figure 5D–F, H*). Thus, Toll-2 signalling can promote G1 to S transition in the pupal brain. To test whether Toll-2 could promote cell cycling also in the adult brain, we heat-shocked flies at the adult critical period. PCNA-GFP+ cells were seen in control adult brains, and over-expression of *Toll-2* increased their incidence (*Figure 5G,H*). However, increases were not statistically significant. Perhaps not all adult progenitors were in G1, as with PCNA-GFP those in G2 would have been missed, and perhaps Toll-2 also provoked cycling from S-phase to G2 or to G2 to mitosis (M).

To test whether there might be progenitors in G2 in the adult brain, we visualised Cdc25/String (Stg), which is expressed in G2 and triggers G2/M transition and entry into mitosis (*Edgar and O'Farrell, 1989*). Using flies bearing a Stg-GFP fusion protein (*Buszczak et al., 2007*), we found Stg-GFP+ cells throughout the adult brain, both in controls and in brains over-expressing *Toll-2* (*Figure 5I*). These data showed that there are cycling cells in G2 or G2/M in the adult brain.

To test whether Toll-2 could induce cell cycling from progenitors arrested in either G1 or G2, we used Fly-FUCCI, at the adult critical period. FUCCI reveals cycling cells in G1, G1/S, G2 and G2/M phases, but not non-cycling cells in G0 (*Zielke et al., 2014*). It drives expression of degron fusion proteins to cell cycle factors E2F and cyclin-B, which get destroyed as cells enter S-phase or G1, respectively. Cells that are only E2F-GFP+ are in G1, cells that are only CycB-RFP+ are in S phase, and cells that are both E2F-GFP+ and CycB-RFP+ are in G2 or M. Over-expressing FUCCI at the adult critical period only, we found cells that were E2F-GFP+ (G1), some that were CycB-RFP+ (S) and some that were both E2F-GFP+ and CycB-RFP+ (G2/M) (*Figure 6A–C*). This meant that the normal adult brain bears cycling cells, presumably progenitor cells resting in G1 or G2. Over-expression

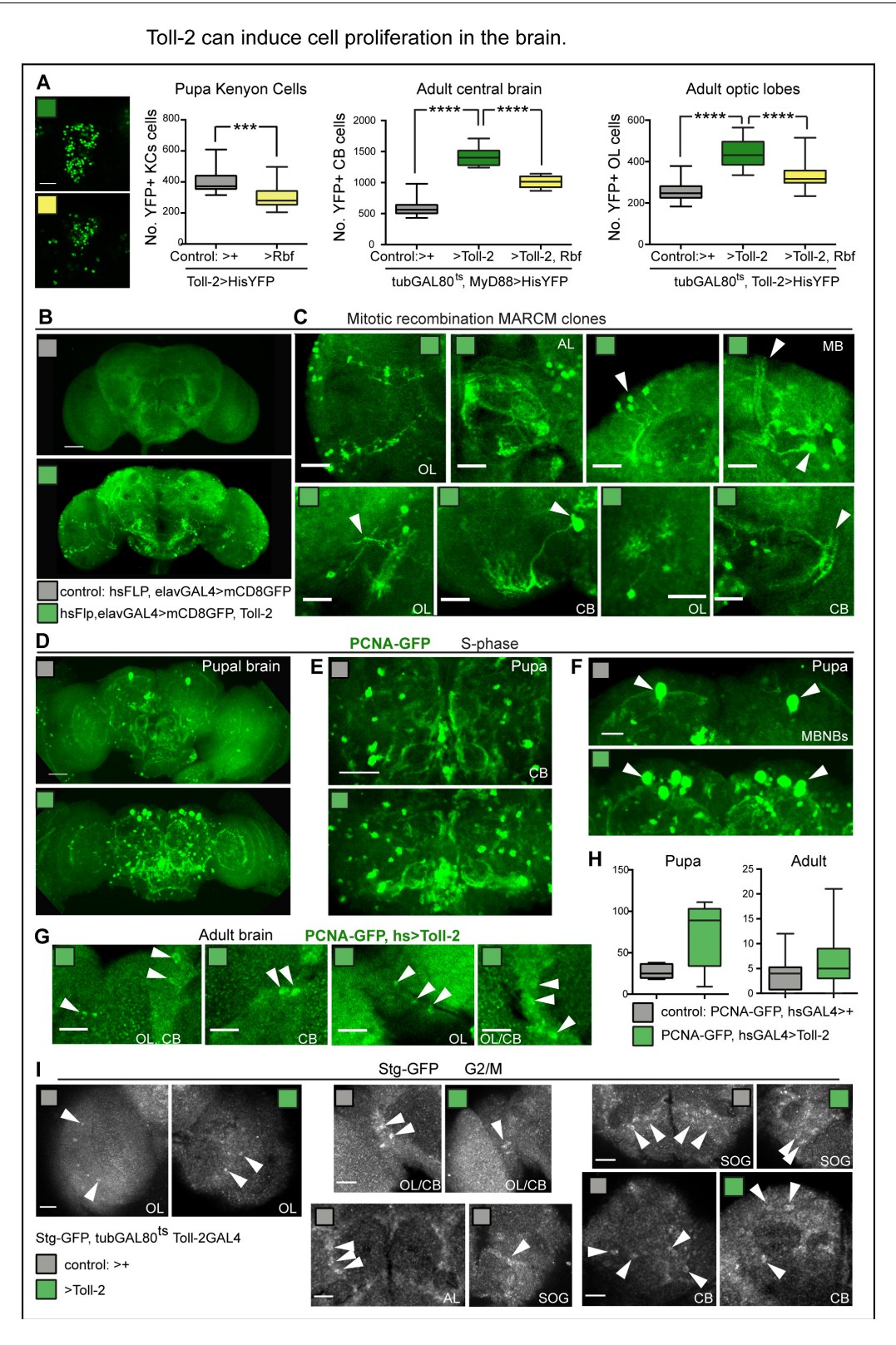

**Figure 5.** Toll-2 can induce cell proliferation in the brain. (**A**) Toll-2 induces G1/S cycling. Over-expression of the G1/S inhibitor *Rbf[280]* in Toll-2+ cells reduced Toll-2 >his-YFP+ KC cell number by day one pupa, compared to controls, meaning that Toll-2+ is required for MB neuroblast divisions. Left box-plot graph: Mann Whitney U test p=0.0004. Middle graph: Conditional over-expression of *Rbf[280]* with *tubGAL80[ts], MyD88GAL4* at the adult critical period rescued the increase in cell number caused by *Toll-2* gain of function, in central brain and optic lobes, meaning that Toll-2 induced cell division.

*Figure 5 continued on next page*

*Figure 5 continued*

Middle and right box-plot graphs: One Way ANOVA p<0.0001, and post-hoc Bonferroni multiple comparisons corrections. (B,C) Toll-2 induced cell division. Mitotic recombination MARCM clones: no clones were found in control brains (B, n = 17), whereas over-expressing *Toll-2* only at the adult critical period resulted in ectopic neurons with differentiated neurites projecting in multiple brain domains (B, C n = 4/17 brains). (D–H) Toll-2 induces G1/S cycling: Conditional over-expression of *Toll-2* with *hsGAL4* increased the number of cells with the S-phase marker PCNA-GFP in pupal brains, compared to controls, albeit not significantly. (E,F) Note particularly more PCNA-GFP+ cells in central brain and mushroom body neuroblasts. (G) PCNA-GFP+ cells were found in adult control brains, and increased with *Toll-2* gain of function. (H) Box-plots, quantification showed increases, albeit not statistically significant. (I) Stg-GFP fusion protein revealed cells in G2/M in both control brains and brains over-expressing *Toll-2* (arrowheads). AL: antennal lobe; OL: optic lobe; CB: central brain; SOG: sub-esophageal ganglion. Scale bars: A,B,D,E: 50 μm; C,F,G,I: 25 μm. For sample sizes, genotypes and statistical details, see *Supplementary file 2*. *p<0.05; ***p<0.001; ****p<0.0001.

of *Toll-2* together with *FUCCI* at the adult critical period, increased the number of cells in G1, S and G2/M phases of the cell cycle compared to controls (*Figure 6A–D*). This means that there are progenitor cells in the adult brain, and Toll-2 can induce their cycling.

To conclude, the above data showed that Toll-2 can induce cell cycling and proliferation in the developing pupal brain and in the adult brain. Furthermore, data indicated that this involved Toll-2 repressing the MyD88 pathway and activating a Wek pathway downstream.

## Toll-2 promotes cell cycling in the brain via yorkie

The finding that over-expression of *Toll-2* or *wek* increased cell number, even further if *MyD88* was also knocked-down, suggested that Wek and MyD88 antagonise each other to regulate cell cycling downstream of Toll-2 in the adult brain. Progenitor cells had been previously reported in the *Drosophila* adult brain (*Kato et al., 2009*; *Fernández-Hernández et al., 2013*; *Foo et al., 2017*). Thus, we revisited the identity of Toll-2+ and MyD88+ cells, and asked whether they included progenitor cells.

In the adult brain, most Toll-2 >hisYFP+ were Elav+ positive neurons, and none were Repo+ glia, but there were some Toll-2 >hisYFP+ Elav-negative and Repo-negative cells, in optic lobes and central brain (*Figure 6E,F*). Amongst them, were large Toll-2+ and MyD88+ cells (*Figure 6G,H*). The large MyD88 >HisYFP+ cells were never Elav+, and some but not all were Repo+ (*Figure 6G*). Using neuroblast markers anti-Miranda (Mira) and anti-Dpn, we found labelled cells in the normal adult brain, as previously reported (*Fernández-Hernández et al., 2013*; *Foo et al., 2017*). Many MyD88 >hisYFP+ cells in the central brain and optic lobes were also Dpn+ (*Figure 6I*), and some MyD88 >hisYFP+ cells also had Mira (*Figure 6J*). Most prominently, many of the large MyD88 >hisYFP+ cells were Dpn+ (*Figure 6I*). Altogether, these data showed that Toll-2 and MyD88 are expressed in progenitor cells in the adult brain at the critical period. To test whether these MyD88+ progenitor cells coincided with those revealed by PCNA-GFP, Stg-GFP and FUCCI, and identify their cell cycle state, we stained *MyD88 >FUCCI* cells with anti-Dpn. Upon *Toll-2* over-expression in the adult, some Dpn+ cells were GFP—RFP+ cells (i.e. in S-phase), some GFP+ RFP+ (i.e. in G2/M), and many were GFP+ RFP— (i.e. in G1)(*Figure 6K*). Thus, as *MyD88* knock-down increased cell number, this meant that MyD88 is expressed in adult progenitor cells, where it normally prevents cell cycling. Toll-2 can overcome this repression.

Then, how does Toll-2 regulate cell cycling in the adult brain? Yorkie (Yki) is a positive regulator of cell proliferation, and it functions downstream of Toll-1 both in immunity and cell competition (*Koontz et al., 2013*; *Liu et al., 2016a*; *Katsukawa et al., 2018*). Yki can regulate the G1-S cyclin *cycE*, but is best known for regulating G2-M *stg* and entry into mitosis, for which Yki shuttles highly dynamically in and out of the nucleus (*Huang et al., 2005*; *Manning et al., 2018*). To test whether cell proliferation at the adult critical period might involve Yki, and whether *yki* could be a target of Toll-2 signalling, we first visualised Yki in the brain. Using a Yki-GFP protein fusion (*Fletcher et al., 2018*), we found many Yki-GFP+ cell nuclei in the normal, adult brain at the critical period (*Figure 7A–C*). Conditional over-expression of *Toll-2* at the critical period increased Yki-GFP+ nuclei throughout the brain (*Figure 7C,D*). Yki-GFP+ nuclei were numerous, but because of cytoplasmic signal in other cells, automatic cell counting with DeadEasy was not possible. Thus, to reliably count Yki-GFP+ nuclei manually, we focused on three regions of interest (ROI): the optic lobe medulla; the sub-esophageal ganglion (SOG) and a top left anterior corner in the central brain (CB). There were Yki-GFP+ nuclei in control brains, that resembled the MyD88+ cells, most noticeably in the optic

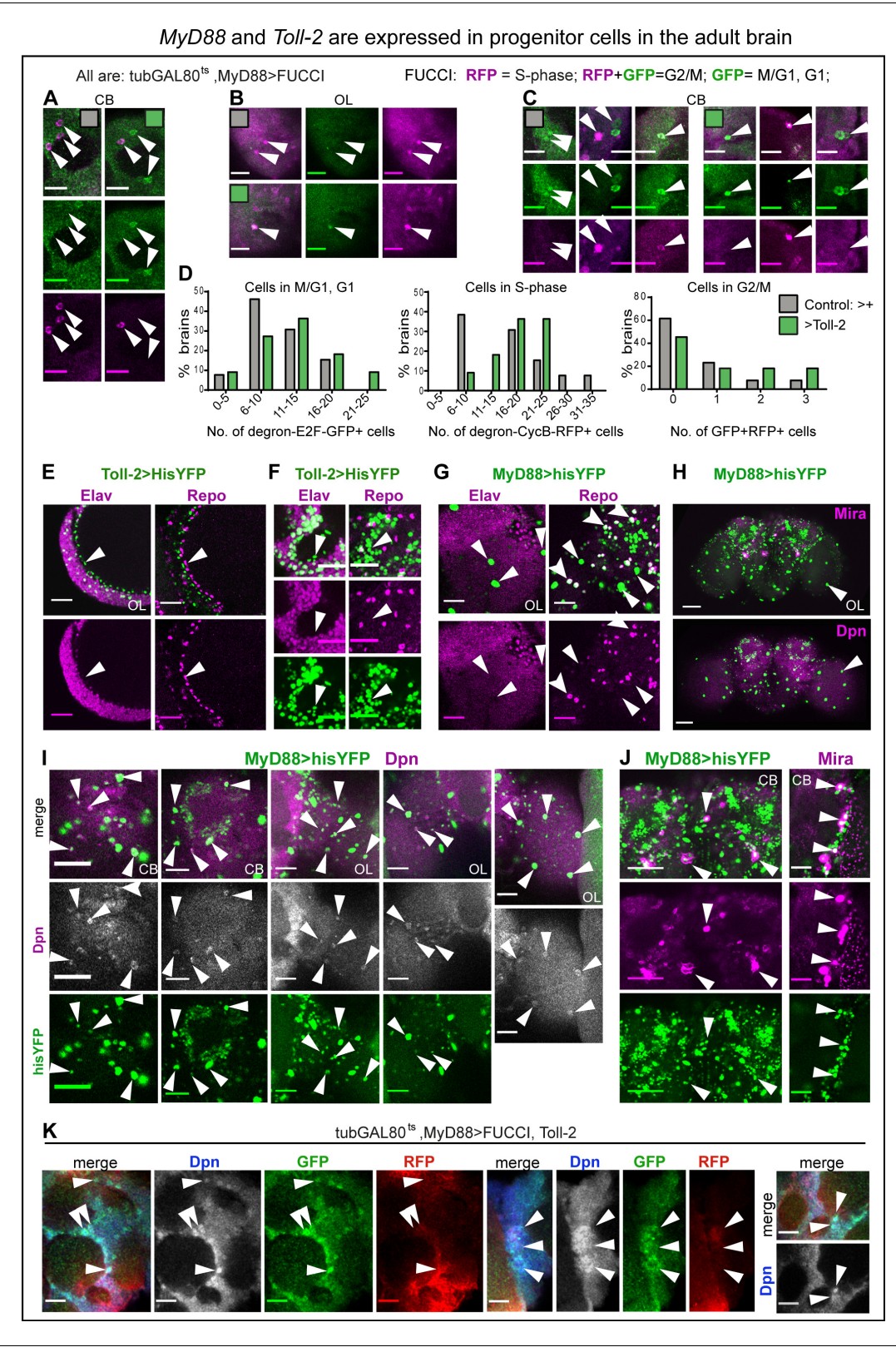

**Figure 6.** MyD88 and Toll-2 are expressed in progenitor cells in the adult brain. (**A–D, K**) FUCCI expressed with *tubGAL80^ts, MyD88GAL4* revealed cycling cells in the adult brain. Arrows point to GFP— RFP+ cells in S-phase, GFP+RFP— cells in G1, and GFP+RFP+ cells in G2/M. (**D**) Quantification: histograms showing that *Toll-2* over-expression increased the number of cells in G1, S- and G2/M phases of the cell cycle. (**E,F**) Some Toll-2^pTV>hisYFP+ cells in optic lobes (**E**) and central brain (**F**) lack pan-neuronal Elav and pan-glial Repo markers (arrows), meaning they are neither neurons

*Figure 6 continued on next page*

*Figure 6 continued*

nor glia. (**G**) Some MyD88 >hisYFP+ cells lack both Repo and Elav markers (arrows), meaning they are neither neurons nor glia. Some but not all large MyD88 >hisYFP+ cells in optic lobes are Repo+. (**H,I**) Co-localisation of MyD88 >hisYFP with the neuroblast marker anti-Dpn, in central brain and optic lobes (arrowheads). Notice the large MyD88+ cells in optic lobes (**H,I**), many of which are Dpn+ (**I**, arrowheads). (**H,J**) Co-localisation of MyD88 >hisYFP with the neuroblast marker anti-Mira in central brain (arrowheads). Mira also seems to label neurons in adult brains. (**K**) MyD88+ progenitor cells cycle in the adult brain. Co-localisation of tubGAL80$^{ts}$, MyD88 >FUCCI with anti-Dpn in adult brains over-expressing *Toll-2*. Some Dpn+ cells over-expressing *Toll-2* were GFP+RFP+ meaning they were in G2/M, and most were GFP+RFP— meaning they were in G1, either just exited mitosis or quiescent. For genotypes, sample sizes and statistical details, see ***Supplementary file 2***. \*\*\*\*p<0.0001. Scale bars: A-G, - I,K : 25 μm; H, J, : 50 μm.

lobes (***Figure 7A,B***). Over-expression of *Toll-2* altered the distribution of Yki-GFP+ nuclei: in the medulla, most brains had fewer Yki-GFP+ nuclei than controls, but 20% brains had more; in the SOG and CB, over-expression of *Toll-2* resulted in more Yki-GFP+ nuclei (***Figure 7C,D***). Both the bimodal distribution and increase in YFP+ nuclei, mean that over-expression of *Toll-2* induced cell cycling, which provoked shuttling of Yki both in and out of the nucleus. Thus, these data showed that nuclear Yki is present in the adult brain at the critical period, and its shuttling is regulated by Toll-2.

Yki shuttles into the nucleus to promote entry into mitosis, and *stg* is one of its targets (***Huang et al., 2005***; ***Manning et al., 2018***). We showed that *Toll-2* overexpression increased cell number, Stg and the incidence of nuclear Yki-GFP in the adult brain. Thus, to further test whether Toll-2 might induce proliferation via Yki in the brain, we used genetic epistasis. We asked whether the increase in MyD88+ cell number caused by over-expression of *Toll-2* in the central brain (*tubGAL80$^{ts}$, MyD88 >Toll-2*) could be rescued by knocking-down *yki*. And it did: conditional *yki-RNAi* knock-down rescued the excess of cell number caused by *Toll-2* gain of function, restoring cell number to control levels in the central brain (***Figure 7E,E'***). Thus, Yki functions downstream of Toll-2 to increase cell number in the adult brain.

Since both over-expression of *wek* and knock-down of *MyD88* increased cell number, we asked whether Wek might regulate cell number downstream of Toll-2, to antagonise MyD88. We tested this using genetic epistasis. Indeed, *wek* RNAi knock-down in MyD88+ cells rescued the increase in cell number caused by *Toll-2* over-expression (***Figure 7F***), demonstrating that Toll-2 increases cell number via Wek.

To conclude, there are quiescent MyD88+ progenitors in the adult brain. MyD88 prevents their proliferation and promotes quiescence, whereas Toll-2 signaling via Wek overcomes their repression to induce proliferation, which requires Yki (***Figure 7G***).

## Tolls regulate brain size in development and in the adult critical period

The above data showed that Toll-2 can regulate cell survival and cell proliferation in the developing and adult brain. There are nine Tolls, and most of them are expressed in the brain. Thus, redundancy between the Tolls might obscure the effects of altering the levels of only one. Different Tolls could also bring about different cellular outcomes - for instance, Toll-1 is more pro-apoptotic than Toll-6 and −7 in pupa (***Foldi et al., 2017***) – compounding the phenotypes. Importantly, the spatial, distinct expression patterns of Tolls could fine-tune the size of distinct anatomical domains. Thus, we asked what effect might down-regulating multiple *Tolls* at once have in the brain. We tested two different RNAi lines for each Toll. Upon sustained knock-down using *Toll-2$^{pTV}$GAL4*, all combinations except one resulted in pupal lethality, precluding analysis in the adult brain. Thus, we analysed pupal brains. Simultaneous knock-down of *Toll-2,–7, −1* or *Toll-2,–7, −8*, or *Toll-2,–7, −6* with *Toll-2$^{pTV}$GAL4* resulted in smaller brains than controls (***Figure 8A***). There was noticeable loss of cells in the optic lobes of *Toll-2$^{pTV}$GAL4 > Toll-2,–7, −8-RNAi*, and *Toll-2,–7, −6-RNAi* (***Figure 8A***). Cell density varied across modules, rendering automatic cell counting with DeadEasy inaccurate, thus we measured brain size instead. Overall brains were smaller in all these genotypes compared to controls, with *Toll-2$^{pTV}$GAL4 > Toll-2,–7,−6-RNAi* having the smallest brains (***Figure 8A,B***). Optic lobe, central brain and KC cluster area were all reduced compared to controls (***Figure 8A,B*** and ***Figure 8—figure supplement 1A,B***). Knock-down of multiple *Tolls* resulted in deeper KC clusters along the A/P axis (***Figure 8D***), revealing that KC clusters were disorganized. In *Toll-2$^{pTV}$GAL4 > Toll-2,–7 −6-RNAi* brains, the reduction in KC cluster area also correlated with a decrease in KC number (***Figure 8C, E*** and ***Figure 8—figure supplement 1B***). Altogether, these data showed that during development: (1) down-regulation of multiple Tolls disrupts brain structure and reduces brain size. (2) Multiple Tolls

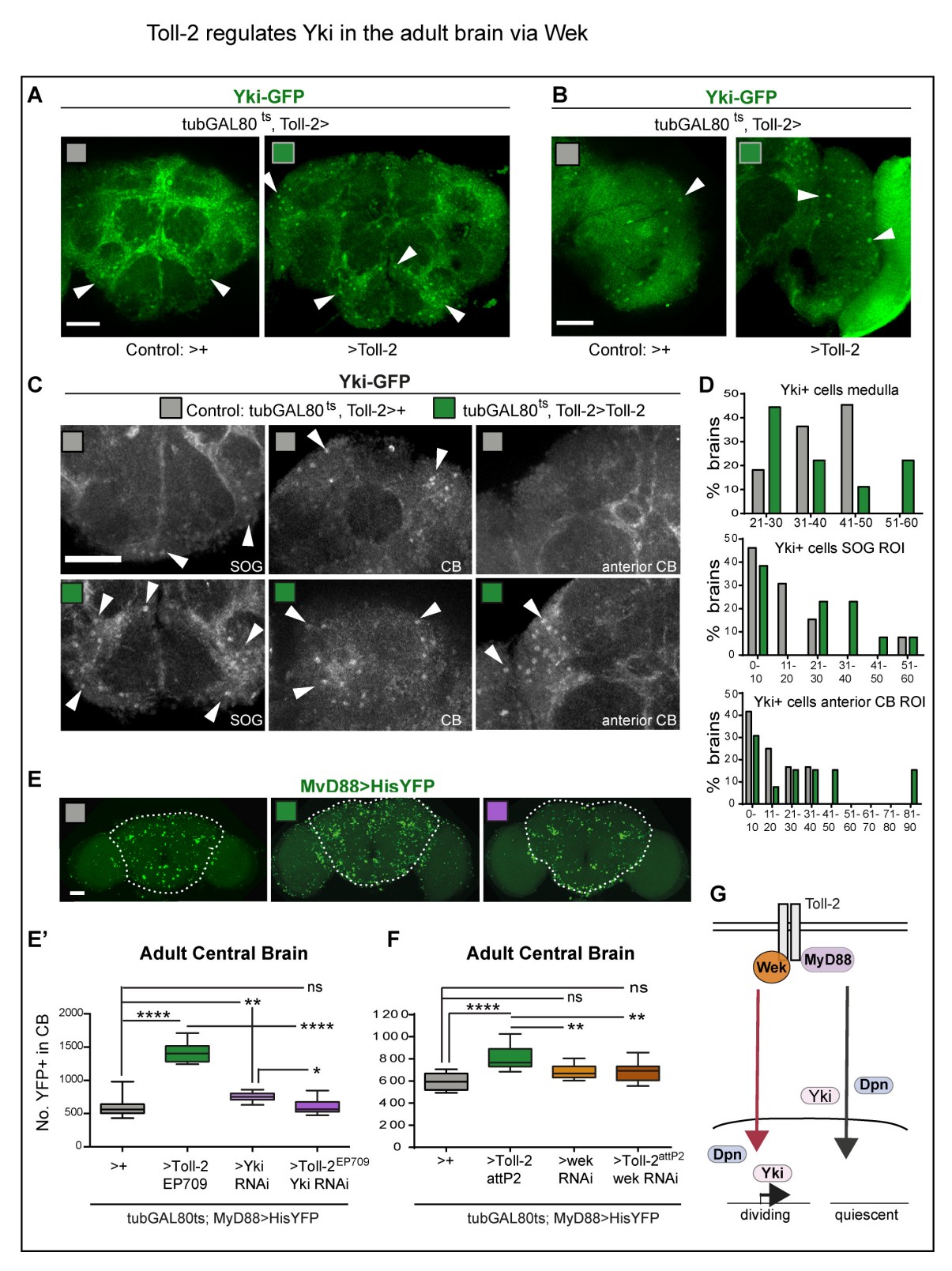

**Figure 7.** Toll-2 regulates Yki in the adult brain. (A–D) Nuclear translocation of a Yki-GFP fusion protein at the adult brain critical period: (A) there were many cells with nuclear Yki-GFP in the adult brain, particularly upon conditional *Tolll-2* over-expression in Toll-2+ cells *(tubGAL80^ts, Toll-2GAL4 > Toll-2)*. (B) In the optic lobes, Yki-GFP+ cells resembled the large MyD88+ cells (see ***Figure 6H***). (C) *Toll-2* over-expressing brains had more Yki-GFP+ cells (arrows). SOG: sub-esophageal ganglion; CB: central brain. Images show (B) medulla and (C) anterior CB and SOG ROIs used for cell counting. (D)

*Figure 7 continued on next page*

Figure 7 continued

Quantification: histograms showing that *Toll-2* over-expression changed the distribution to a bimodal profile in medulla as some brains had fewer Yki-GFP+ cells and some had more, and increased cell number in the SOG and CB, meaning that Toll-2 induced cell cycling and shuttling of Yki-GFP. (E,E') The increase in MyD88 >hisYFP+ cell number in the central brain caused by *Toll-2* over-expression was rescued with *yki* RNAi knock-down. Box-plots, One Way ANOVA p<0.0001, and post-hoc Bonferroni test. (F) The increase in MyD88 >hisYFP+ cell number in the central brain caused by *Toll-2* over-expression was rescued with *wek* RNAi knock-down. Box-plots, One Way ANOVA p<0.0001, and post-hoc Bonferroni test. (G) Diagram illustrating that Toll-2 can prevent progenitor cycling via MyD88, and activate it via Wek and Yki. Scale bar in (A,B,C,E): 50 μ.*p<0.05, **p<0.01, ***p<0.001, ****p<0.0001. For all genotypes, sample sizes and statistical details, see *Supplementary file 2*.

regulate cell number in the optic lobes and central brain. (3) Tolls function in concert to regulate KC cluster size, organisation and KC number.

Next, we asked whether conditional knock-down of multiple Tolls might affect the adult brain. For this, due to genetics limitations, we could only test two *Tolls* at a time. Conditional knock-down of both *Toll-2* and −6 with *MyD88GAL4,* restricted to the adult critical period, increased cell number in the central brain (*Figure 8—figure supplement 1C,D*). This was reminiscent of the increase in cell number caused by *MyD88* knockdown (*Figure 4E*), and could also involve the pro-apoptotic function of Toll-6 (*Foldi et al., 2017*). *Toll-2^{pTV}GAL4* flies are heterozygous mutant for *Toll-2, and together with RNAi knock-down of Toll-6,−7 with Toll-2GAL4* did not affect brain size, but with *Toll-2,−6* knock-down most brains were smaller than controls, and around 25% were bigger (*Figure 8—figure supplement 1E,F*). Most remarkably, whereas conditional manipulation of *Toll-2* alone did not affect KCs (*Figure 4B,F*), conditional knock-down of both *Toll-2,−6* or *Toll-6,−7* in the adult altered KCs (*Figure 8F–H*). KC number and cluster depth increased with *Toll-6,−7-RNAi,* and decreased slightly (albeit not significantly) with *Toll-2,−6-RNAi* knock-down (*Figure 8F–H*). These data meant that: (1) different Tolls have distinct functions at the adult critical period; (2) Tolls have distinct as well as redundant or overlapping functions in KCs; and (3) KC organization and number can be altered in the adult by Tolls.

Experience increases brain size by around 5% compared to un-stimulated controls (*Heisenberg et al., 1995*; *Barth and Heisenberg, 1997*; *Barth et al., 1997*). Thus, in view of the above results, we asked whether conditional manipulation of *Toll-2* or its signalling pathways could influence brain size at the adult critical period. Most remarkably, over-expression of *Toll-2* in MyD88 + cells at the adult critical period (*tubGAL80^{ts}, MyD88 >Toll-2*) increased brain size compared to controls (*Figure 8I,J*). Even if only female brains were analysed, there was considerable variability in brain size in controls, precluding statistical significance (*Figure 8J*, grey box-plots). Still, both the median (box-plots in *Figure 8J*) and mean differed: *Toll-2* over-expressing brains were on average 5.4% (using UAS-Toll-2^{EP709}) and 7.4% (using UAS-Toll-2^{attP2}) larger than control brains, respectively (i.e. mean control brain size was 138,585μ$^2$ vs. 146,177μ$^2$ of *tubGAL80^{ts}, MyD88 >Toll-2^{EP709}* and 148,861μ$^2$ of *tubGAL80^{ts}, MyD88 >Toll-2^{attP2}* brains). Furthermore, whereas 40% of control brains were larger than the mean, around 80% of *Toll-2* over-expressing brains were larger than the control mean (>140,000) (*Figure 8K*, left). Conditional *MyD88-RNAi* (e.g. *tubGAL80^{ts}, MyD88 >MyD88* RNAi) knock-down or over-expression of *wek* resulted in even larger brains, with virtually all brains being larger than both the control median and mean (14.7% larger for *wek* and 20% for *MyD88-RNAi*, *Figure 8I,J,K*). These results were consistent with both MyD88 promoting quiescence and Wek promoting cell proliferation. Surprisingly, however, although the concerted over-expression of *wek* and *MyD88-RNAi* increased cell number (*Figure 4E*), it did not increase brain size (*Figure 8J,K*), perhaps because together they can also induce apoptosis (*Foldi et al., 2017*). Whereas conditional *yki-RNAi* knock-down in MyD88+ cells (*tubGAL80^{ts}, MyD88 >yki RNAi*) resulted in highly variable brain size, *yki-RNAi* could rescue the increase in brain size caused by *Toll-2* gain of function (*Figure 8J,K*). This was consistent with Yki functioning downstream of Toll-2 to induce cell proliferation in the adult brain. To conclude, these data showed that, like experience, Toll-2 can positively regulate brain size at the adult critical period. This involved counteracting MyD88 and activating Wek and Yki signalling downstream.

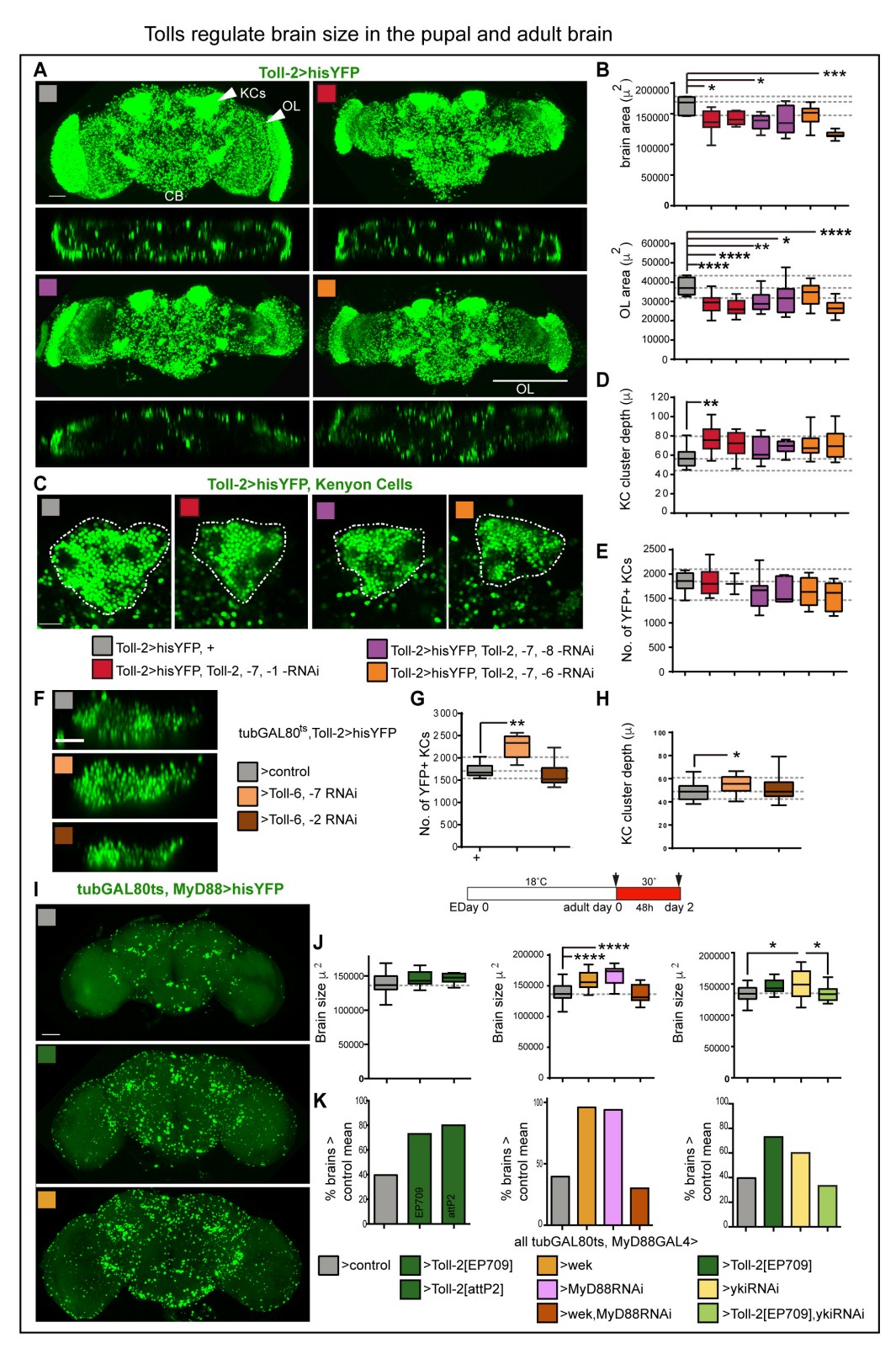

**Figure 8.** Tolls regulate brain size in the pupal and adult brain. (**A,B**) Constant RNAi knock-down of three *Tolls* with *Toll-2GAL4* resulted in smaller brains. Optic lobes were smaller than in controls, and so were central brains (*Figure 8—figure supplement 1A*). (**B**) Brain area box-plots: both are One Wat ANOVA p<0.01, post-hoc Dunnett test comparisons to control. (**C,D**) In the triple *Toll* RNAi knock-out, Kenyon cell clusters were disorganized, with smaller area (*Figure 8—figure supplement 1B*) whilst deeper along the A/P axis, and (**E**) had fewer cells (albeit not significantly). (**D**) Box-plots, Kruskal-

*Figure 8 continued on next page*

*Figure 8 continued*

Wallis ANOVA, p=0.0127, post-hoc Dunn. (F–H) Conditional knock-down of two *Tolls* with *tubGAL80^ts*, *Toll-2GAL4* at the adult critical period disorganized Kenyon cell clusters and altered Kenyon cell number: *Toll-2,–6* RNAi reduced cell number, whereas *Toll-6 ,–7* RNAi increased KC depth and cell number. (G) Box-plots, Kruskal Wallis ANOVA p<0.0001, post-doc Dunn. (H) Box-plots, One Way ANOVA p=0.0567, post-hoc Dunnett. (I–K) Conditional alteration of *Toll-2* function with *tubGAL80^ts*, *MyD88GAL4* at the adult critical period (day 0–2) increased brain size: (J) Left: over-expression of *Toll-2* increased brain size, using two different *UAS-Toll-2* lines: *EP709* (2^nd chromosome) and an insertion into *attP2* (3^rd chromosome). Middle: both over-expression of *wek* and *MyD88* RNAi increased brain size, but over-expression of both did not, possibly because this genotype can also induce cell death. Right: over-expression of *yki*-RNAi alone resulted in a variable phenotype, but rescued the increase in brain size caused by *Toll-2* over-expression. Box-plots, dashed lines indicate median of controls: left, One Way ANOVA, p=0.0793, not significant; centre, Kruskal Wallis ANOVA p<0.0001, post-hoc Dunn; right, One Way ANOVA p<0.05, post-hoc Tukey. (K) Mean increases in brain size ranged from 5.4% to 20% compared to controls, but most remarkably there was a persistent increase in the percentage of brains larger than the control mean of 138,585μ² upon *MyD88* RNAi knock-down, over-expression of *Toll-2* or *wek. yki* RNAi knockdown fully rescued the *Toll-2* gain of function phenotype. Scale bar in (A,I): 50 μ; (C,F): 25 μ.*p<0.05, **p<0.01, ***p<0.001, ****p<0.0001. Two different RNAi lines were used for each *Toll*. (B,D,E,G,H) Dashed lines indicate median, maximum and minimum values in controls. For genotypes, sample sizes and statistical details, see *Supplementary file 2*. See *Figure 8—figure supplement 1*. The online version of this article includes the following figure supplement(s) for figure 8:

**Figure supplement 1.** Conditional knock-down of *Toll-2 and −6* in the adult brain altered cell number and brain size.

## Discussion

At least seven of the nine *Toll*-receptors are expressed topographically, mapping the distinct modules that form the brain. Toll receptors regulate cell number and brain size in development and structural brain plasticity in the adult, through their ability to promote either cell survival or death, progenitor cell quiescence or proliferation. Evidence indicates that Tolls can underlie the changes that experience brings about in the adult brain (*Technau, 1984*; *Heisenberg et al., 1995*; *Barth and Heisenberg, 1997*; *Barth et al., 1997*), and that structural plasticity and neurodegeneration are two faces of Toll-driven cellular responses in the brain (*Figure 9*).

Toll-2 promotes neuronal survival and proliferation, both in development and in the adult brain. *Toll-2* is neuroprotective as loss of function caused neurodegeneration: it increased apoptosis and caused neuronal loss, and *Toll-2* mutant neurons that survived had dendrite loss, axon atrophy and misrouting. *Toll-2* loss of function also impaired climbing and walking, and decreased lifespan, phenotypes characteristic of neurodegeneration. Toll-2 promotes cell survival through the canonical MyD88-NFκB pathway, as previously found for the pro-survival functions of Toll-6 and −7 in development (*McIlroy et al., 2013*; *Foldi et al., 2017*). Both in flies and mammals, Tolls and TLRs promote cell survival via MyD88 and cell death via Sarm, which activates the pro-apoptotic function of JNK and inhibits MyD88 (*Kim et al., 2007*; *McIlroy et al., 2013*; *Mukherjee et al., 2015*; *Foldi et al., 2017*). Distinct Tolls and TLRs can preferentially promote cell survival or cell death, as for instance, Toll-1 is more pro-apoptotic than Toll-6 (*Foldi et al., 2017*), and in mammals TLR4 promotes neuronal survival and TLR8 neuronal death (*Ma et al., 2006*; *Zhu et al., 2016*). We showed that *Toll-2* over-expression did not cause cell loss, meaning that Toll-2 is not pro-apoptotic in the brain. Altogether, data showed that Toll-2 is neuroprotective in the brain.

A novel molecular pathway underlies the ability of Toll-2 to regulate cell proliferation in development and structural brain plasticity in the adult. A remarkable finding was that *Toll-2* gain of function in the pupal or adult brain not only maintained neurons alive, but also induced cell cycling - which was visualized with standard cell proliferation markers PCNA-GFP for S-phase, FUCCI for G1, G2, G2/M, and Stg and nuclear Yki for G2/M and M phases -, it increased cell number and brain size. We showed that there are progenitor cells in the adult brain that are kept quiescent by MyD88, and loss of MyD88 at the adult critical period increased neuronal number and brain size. As MyD88 is the general adaptor for canonical Toll signalling, this implies that Toll signalling maintains progenitor cells quiescent. Cell proliferation was induced when the repression by MyD88 was overcome by *Toll-2* over-expression and signalling via the alternative adaptor, Wek. Conditional over-expression of *Toll-2* or *wek,* or knock-down of *MyD88*, at the adult critical period increased neuronal number and brain size. Furthermore, the effect of Toll-2 was dependent on both Wek and Yki, a well-known inducer of cell proliferation (*Koontz et al., 2013*). Over-expression of *Toll-2* induced nuclear translocation and shuttling of Yki, correlating with nuclear localization of Stg, target of Yki, in the brain. Furthermore, genetic epistasis analysis showed that the increase in cell number caused by *Toll-2* over-expression could be rescued with either *yki-RNAi* or *wek-RNAi* knockdown. Thus, Toll-receptor

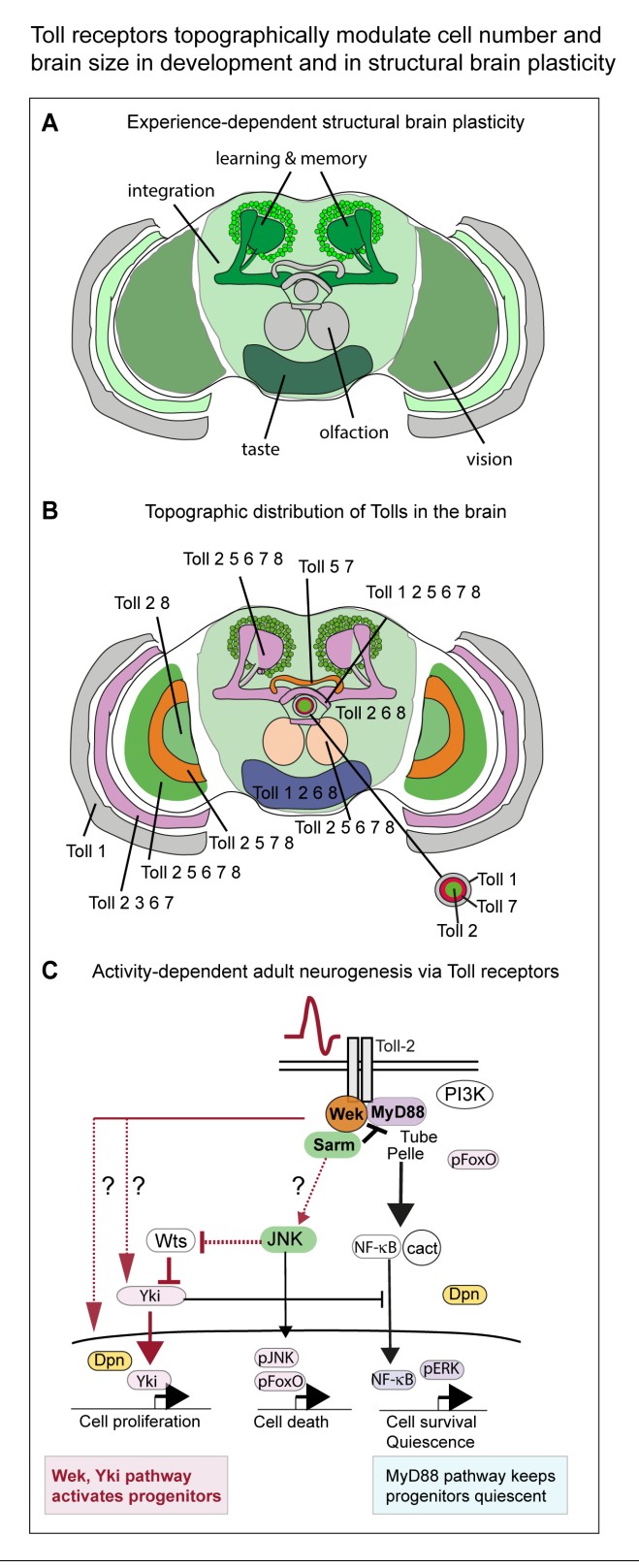

**Figure 9.** Toll receptors topographically modulate experience-dependent structural brain plasticity. (**A**) Experience – that is stimulation through vision - alters the size of multiple brain domains, including optic lobes (for vision), central brain (integration, equivalent to mammalian cortex) and Kenyon cells (learning and memory). (**B**) Topographic distribution of Tolls in the brain reveals: (1) a map of *Toll* expression profiles coincident with

*Figure 9 continued on next page*

*Figure 9 continued*

anatomical brain domains; (2) profiles specific to each *Toll*; (3) complementary patterns in neuropiles of the visual system and central complex (fan shaped body, ellipsoid body); (4) overlapping distributions in optic lobes, protocerebral bridge, Kenyon cells and mushroom bodies. Each brain module and neuropile expresses a different *Toll* or combination of *Tolls*, which can potentially regulate each region differentially in development and in the adult brain. (C) Activity-dependent adult neurogenesis via Toll receptors. Neuronal activation at the adult critical period increased cell number through a Toll-2-dependent mechanism. Toll receptors can signal via MyD88 or Wek in the adult brain. Via MyD88, they maintain adult progenitor cells quiescent, and thus repress cell proliferation. Via Wek, they promote progenitor cell cycling and proliferation, which also requires Yki. How Wek relates to Yki has not been solved (question marks indicate conceivable alternatives). Conditional over-expression of *wek* restricted to the adult critical period increases brain size. Thus, Tolls can promote cell proliferation in the adult brain through a novel mechanism involving Wek and Yki, that antagonizes the function of MyD88 in promoting quiescence.

signalling can switch between promoting quiescence via MyD88 to promoting cell proliferation via Wek. Whether Wek might activate cell proliferation directly, or activate Yki or JNK first, was not solved. Wek also induces cell death, by linking Tolls to Sarm, which activates pro-apoptotic JNK signalling (*Foldi et al., 2017*). JNK can also induce cell proliferation, and Yki is activated downstream of JNK and Toll in other contexts (*Enomoto et al., 2015*; *Gerlach et al., 2018*; *Katsukawa et al., 2018*). Thus, Wek could activate cell proliferation directly and independently of Yki, or it could activate Yki either directly, or via Sarm and/or JNK. Any of these options is possible, as in immunity and in cell competition (which also involves regulation of cell number), Toll-1 can regulate Yki downstream via JNK-dependent and independent pathways (*Liu et al., 2016a*; *Katsukawa et al., 2018*). Either way, our data showed that Toll-2 can prevent or induce progenitor cell proliferation, through alternative MyD88 or Wek signalling pathways downstream (*Figure 9C*).

Knock-down of multiple *Tolls* through development severely altered brain structure and reduced brain size. Most likely, Tolls promote either cell survival or cell proliferation or both during brain development, and together they modulate brain formation. How each of them may influence the adult brain, is more difficult to dissect. Through their ability to elicit multiple cellular outcomes, Tolls can have distinct, redundant, synergistic, antagonistic or compensatory functions. For instance, whereas Toll-1 and −6 can have pro-apoptotic functions in the pupal VNC (*Foldi et al., 2017*), Toll-2 was not pro-apoptotic in the brain. Altering *Toll-2* function alone did not affect Kenyon Cells, but simultaneous persistent knock-down of three *Tolls* reduced KC number and disorganized KC clusters; and whereas conditional knock-down of *Toll-2,–6* at the adult critical period reduced KC number, conditional knock-down of *Toll-6,–7* increased KC number. Thus, although all Tolls could access the same downstream signalling pathways, each Toll modulates these pathways in their own way. As a consequence, knock-down of one or more *Tolls* most likely induced complex responses by other Tolls in the same or neighbouring cells, compounding the phenotypes. What enables Tolls to elicit different cellular outcomes is an intriguing question.

Adult neurogenesis is much debated. Neurogenesis occurs in the adult brain of humans and other animals (*Cayre et al., 1996*; *Deng et al., 2010*), but the extent of it is unknown, and solving this is important to understand how the brain works and brain disease. In *Drosophila*, developmental neural stem cells are eliminated by the end of pupal life (*Ito and Hotta, 1992*; *Maurange et al., 2008*; *Siegrist et al., 2010*; *Hara et al., 2018*). However, neurogenesis has been reported in the adult brain: naturally occurring cell death induces cell proliferation in the adult brain (*Kato et al., 2009*); cell proliferation was reported with mitotic recombination clones (*Fernández-Hernández et al., 2013*); *mir-31* mutations induce glial and neuroblast proliferation in the adult brain (*Foo et al., 2017*); injury induces gliogenesis and neurogenesis (*Kato et al., 2009*; *Fernández-Hernández et al., 2013*; *Moreno et al., 2015*); the partner of Yki, *scalloped*, is expressed in the adult brain (*Rohith and Shyamala, 2017*); immuno-histochemistry and single cell transcriptomics revealed that neuroblast or intermediate neural progenitor (INP) markers *eyeless, dichate, grainy-head, dpn, miranda* and *inscutable* are expressed in the adult brain (*Callaerts et al., 2001*; *Fernández-Hernández et al., 2013*; *Foo et al., 2017*; *Zhu et al., 2017*; *Croset et al., 2018*; *Davie et al., 2018*; *Konstantinides et al., 2018*); and interference with the normal regulation of cell survival and cell death – processes that Tolls can influence - results in ectopic and/or persistent neuroblasts in the

adult brain (*Siegrist et al., 2010*; *Hara et al., 2018*). Consistently with these findings, in the adult brain there are MyD88+ Dpn+ Stg-GFP+ Yki-GFP+ FUCCI+ cells in S-phase or G2/M in the optic lobes, and in S, G1 and G2/M in the central brain. MyD88+ progenitor cells are normally quiescent and Toll-2 gain of function can induce their cell cycling and proliferation at the adult critical period.

Adult progenitor cells may be distinct from developmental neuroblasts. In fact, the fate of quiescent INPs has not been determined, suggesting some could also exist in the adult brain (*Walsh and Doe, 2017*). In other insects, adult progenitor cells differ from developmental neuroblasts, and instead originate from hemocytes (*Simões and Rhiner, 2017*). In the mammalian brain, adult progenitors originate from glia (*Falk and Götz, 2017*). Some of the large Toll-2+ and MyD88+ Dpn+ cells also had the glial marker Repo. Thus, adult progenitor cells may not originate from canonical developmental larval neuroblasts.

Experience alters brain structure topographically, altering the regions involved in experience-dependent processing (*Figure 9A*). For instance, rearing flies in constant light or constant darkness alters the size of brain modules involved in vision (e.g. optic lobes) (*Heisenberg et al., 1995*; *Barth and Heisenberg, 1997*; *Barth et al., 1997*). Neuronal activity increased cell number in the optic lobes in a Toll-2 dependent manner, and conditional over-expression of *Toll-2* or *wek* in the adult brain, at the critical period, increased both cell number and brain size. The anatomical segregation of the seven Toll receptors enables them to modulate cell number within distinct brain modules. This implies that: (1) in development, Tolls could adjust brain neuronal populations topographically to the motor and sensory circuits, enabling appropriate behaviour. (2) In evolution, *Tolls* could have driven changes in brain shape, enabling adaptation to distinct environments and behavioural diversification. (3) In adults, Tolls can enable structural brain plasticity, by adjusting brain neuronal populations topographically to experience-dependent inputs, and drive behavioural adaptation throughout the life-course.

TLRs could operate in analogous ways in the human brain (*Okun et al., 2009*; *Okun et al., 2011*). TLRs are expressed throughout the mammalian brain (*Ma et al., 2006*; *Cameron et al., 2007*; *Rolls et al., 2007*; *Lathia et al., 2008*; *Okun et al., 2009*; *Okun et al., 2010a*; *Okun et al., 2011*; *Okun et al., 2012*). Some TLRs have neuro-protective and other TLRs pro-apoptotic functions, for instance TLR-2 and −4 promote cell survival, and TLR3 and −8 apoptosis, neurite retraction and neurodegeneration (*Ma et al., 2006*; *Hung et al., 2018*). TLRs can regulate neural stem cell proliferation, formation or elimination of neurites and neurons, including during learning and long-term memory (*Ma et al., 2006*; *Rolls et al., 2007*; *Okun et al., 2010a*; *Okun et al., 2012*; *Madar et al., 2015*; *Hung et al., 2018*).

To conclude, through their topographic distribution, Tolls modulate cell number and brain size, in development and in structural plasticity in the adult (*Figure 9*). They do so by engaging different molecular pathways that regulate neuronal survival or death, and progenitor quiescence via MyD88 or proliferation via Wek. It will be compelling to test if the combination of TLR topography and diversity of signalling options downstream, also underlies neurodegeneration and structural plasticity in the human brain.

## Materials and methods
See *Supplementary file 1*: Key Resources Table.

### Genetics
#### Mutants
*Toll-2^{pTV}* is a hypomorphic loss of function allele and *mCherry* knock-in of *Toll-2*, generated via CRISPR/Cas9-meditated homologous recombination and insertion of the *pTV* cassette (*Baena-Lopez et al., 2013*). *Df(2R)BSC594* and *Df(2R)BSC22* rare deficiencies for the *Toll-2* locus (Bloomington Stock Center, BSC). *18w^{D7-35}* is a null *Toll-2* mutant allele (*Eldon et al., 1994*). GAL-4 lines: *Toll-3GAL4^{MI02994}*, *Toll-6GAL4^{MIO2127}* and *Toll-7GAL4^{MI13963}*, were generated here by RMCE from MIMIC insertions into the only exons of these intron-less genes (see below). *Toll-8GAL4^{MD806}* is a P-element insert-180bp upstream of the *Toll-8* start codon. *MyD88GAL4^{NP6394}* is an insert in the 5'UTR,~700 bp upstream of the *MyD88* start codon, and therefore most likely is bona fide reporter of the endogenous *MyD88* expression pattern. *sarm-GAL4^{NP7460}* is a *GAL4* insertion upstream of the four short isoforms; *sarm-GAL4^{NP0257}* is inserted upstream of the longest isoform, and drives expression of all

isoforms. Reporters: (1) Nuclear Dif and Dorsal were visualised in transgenic flies bearing Bacmid clones *Dif-GFP.FPTB* (BSC42673) and *Dl-GFP-FPTB* (BSC42677), with GFP-FLAG tags inserted at C-termini of Dl isoforms A,B,D and E, and Dif isoforms A, B and D, bearing the nuclear internalization signal. (2) *wek-GFP:* w; PBacwek GFP-FLAG-FPTB$^{VK00033}$ (BSC67719) is a CPTI exon-trap insertion into the *wek* locus that generates a Wek-GFP fusion protein. (3) *TRE-RED*, reporter for JNK signaling (gift of Y Fan and BDSC). (4) *UAS-FlyBow1.1* was used without heat-shock to visualise GFP (gift of I Salecker). (5) UAS-myr-td-Tomato is membrane tethered (gift of B Pfeiffer). (6) *UAS-hisYFP* enables nuclear YFP visualization (*Forero et al., 2012*). (6) PCNA-GFP is an S-phase reporter (*Kato et al., 2011*). (7) Yki-GFP is a protein fusion and allows to visualize Yki in and out of the nucleus (gift of B Thompson and N Tapon *Fletcher et al., 2018*). UAS lines: (1) *EP709:* UAS-Toll-2 P-element insertion into the Toll-2 locus. (2) *UAS-TrpA1*, to activate neurons above 27°C. (3) Purposely made *UAS-Toll-2* insertion into *attP2* on third chromosome. (4) *UAS-wek* was published in *Foldi et al. (2017)*. (5) *UAS-Rbf$^{280}$* was published in *Kato et al. (2011)*. (6) *UAS-p35* was published in *Sutcliffe et al. (2013)*. (7) *UAS-Dp110CAAX* drives expression of activated PI3K. RNAi transgenic lines are: UAS-Toll-1RNAi: y[1]v[1], UAS Toll-1-RNAi [P.TRiP.JF10491](BSC31044) and UAS Toll-1 RNAi [TRiP JF01276](BSC31477); UAS-Toll-2RNAi: y[1] sc* v[1]; P{TRiP.HM05241}attP2/TM3, Sb1 (BSC30498), w; UAS Toll-2 RNAi (VDRC V36305) and UAS Toll-2 RNAi (VDRC V44386). UAS-Toll-6RNAi:y[1] v[1]; P{TRiP.HMS04251}attP2 (BSC56048) and w; UAS Toll-6 RNAi (VDRC V928). UAS-Toll-7RNAi: w;UAS Toll-7 RNAi (NIG II)(Kyoto DGRC) and w;;UAS Toll-7 RNAi (NIG III)(Kyoto DGRC). UAS-Toll-8RNAi: yv; UAS Toll-8-RNAi[P.TRiP.HM05005] (BSC28519) and UAS Toll-8 RNAi (VDRCV27098). *UAS-MyD88RNAi[GD9716](VDRC* 25399 and 25402); *UAS-JNK-RNAi[GD10555] (VDRC34138)*; *UAS-cactusRNAi [TRIP.HMS00084]attP2 (BSC34775)*; *UAS-yorkie-RNAi: P{TRIP. JF03119}attP2 (BSC:31965)*; *UAS Wek RNAi [P.TRiP.GLV21045] (BSC35680)*. To select pupae of the desired genotype, the fusion balancer *SM6aTM6B* marked with *Tb-* and which segregates second and third chromosomes together, was used. All experiments were carried out at 25°C unless otherwise indicated.

## Molecular biology, CRISPR/Cas9 gene editing and generation of transgenic flies

*Toll-2$^{pTV}$* flies were generated by CRISPR/Cas9 mediated homologous recombination replacement of the coding region of *Toll-2* for the *pTV* cassette (*Baena-Lopez et al., 2013*). For PCR amplification of genomic regions, primers with ideally no-off targets were designed using the tools in http://www.ncbi.nlm.nih.gov/tools/primer-blast/. Genomic DNA was extracted from Oregon R wild-type flies as Polymerase Chain Reaction (PCR) template. For the 5' homology arm (HA), 4.993 kb upstream of the *Toll-2* start codon were PCR-amplified using primers 3 and 4 (*Supplementary file 3*), and for the 3'homology arm (HA), 3.419 kb downstream if the*Toll-2* CDS starting from the stop codon were amplified by PCR using primers 1 and 2 (*Supplementary file 3*). Conventional ligation cloning was used to insert the 5'HA into *NotI* and *KpnI* enzyme restriction sites in the pTV vector (*Baena-Lopez et al., 2013*), and the 3'HA into *SpeI* and *AscI* sites. Insertion of both 5'HA and 3'HA into the correct locus, as well as the integrity of the pTV cassette were verified by PCR diagnostics. The gRNA targeting Toll-2 was created by annealing primers 5 and 6 in *Supplementary file 3* followed by conventional ligation cloning into the BbsI site of vector pU6.3 (*Port et al., 2014*). Cloning was verified by sequencing. The resulting *Toll-2-gRNA* construct was inserted into the attP2 landing site by FC31 transgenesis. Transgenic flies were crossed to *nos-Cas9* transgenic flies, the *Toll-2pTV* construct was injected into progeny flies (BestGene), and transformants were identified using *w+* as a genetic selection marker. PCR analysis indicates that the *Toll-2$^{pTV}$* insertion deleted half of the coding region of *Toll-2*, including the start codon, generating a mutant allele. *Toll-2$^{pTV}$* carries *mCherry* expressed under the control of the endogenous Toll-2 promoter.

The *GAL4* sequence in the original *pTV* cassette was damaged (*Baena-Lopez et al., 2013*), so *GAL4-attB* (AddGene) was integrated into the *attP*-landing site within the *pTV* cassette, using FC31 transgenesis (injections by BestGene Inc). This placed *GAL4* coding sequences just before the *Toll-2* start codon. UAS-FlyBow was used for genetic selection.

*Toll-4GAL4* and *Toll-5GAL4* were generated by CRISPR/Cas9 facilitated homologous recombination, inserting *T2AGAL4* (AddGene) just upstream of their start codon, whilst retaining intact coding regions, and following the protocol described in *Gratz et al. (2015)*. For the 5' homology arms (HA) and 3'HA, 1 kb fragments were amplified from template genomic DNA from wild-type Oregon R

flies using primers 9–12 (*Supplementary file 3*) for *Toll-4* and primers 15–18 (*Supplementary file 3*) for *Toll-5*, and cloned using conventional ligation cloning into *AgeI* and *NotI* restriction sites for the 5'HAs and into *AscI* and *SpeI* sites for the 3'HAs. *Toll-4* and *Toll-5* gRNAs were generated as above, using primers 13, 14 and 19,20 (*Supplementary file 3*), respectively, and cloned into the *BbsI* site of the *pU6.3* vector. *Toll-4-T2AGAL4* and *Toll-5-T2AGAL4* and *gRNA-pU63* constructs were injected into *nos-Cas9* flies (BestGene Inc) and transformants were identified with *3xP3-DsRed* genetic selection marker, which was subsequently removed by *CRE-Lox* mediated recombination from flanking *LoxP* sites in the *T2A-GAL4 vector*.

*Toll-3GAL4$^{MI02994}$*, *Toll-6GAL4$^{MIO2127}$* and *Toll-7GAL4$^{MI13963}$* were generated by Recombination Mediated Cassette Exchange (RMCE) (*Venken et al., 2011*), by injecting the *GAL4-attB* plasmid into *Toll-3$^{MI02994}$*, *Toll-6$^{MIO2127}$* and *Toll-7$^{MI13963}$* MIMIC lines and using FC31 transgenesis (injections by BestGene Inc). *Toll-3,–6* and *−7* are all intron-less genes, thus GAL4 is expressed under the control of their endogenous promoters.

*UASToll-2* transgenic flies were generated by PCR-amplifying the coding region of the intron-less Toll-2 gene using primers 7,8 (*Supplementary file 3*), followed by insertion into the *pUAS-gw-attB* vector using Gateway cloning and FC31 transgenesis into the *attP2* landing site (BestGene Inc) on the third chromosome. The genetic selection marker *w+* was used to identify transformant flies.

## Reverse transcription PCR

Non-quantitative Reverse Transcription PCR (RT-PCR) was carried out on wild-type Oregon-R embryos, dissected central nervous systems (CNSs) from second and third instar-wandering larvae and pupae, and whole heads. Total RNA was extracted with Trizol (Ambion) and RNA integrity and concentration was confirmed using a Nano-drop. RNA samples were DNase treated to remove residual genomic DNA contamination. 300 ng of RNA was used for cDNA synthesis following GoScript Reverse Transcriptase treatment. Samples were diluted 1:3 with Nuclease free H$_2$0, and a no RT sample of 300 ng of RNA made up to 60 ul with Nuclease free H$_2$0. Standard PCR reaction was performed to amplify each of the Toll receptor cDNA using GoTaq PCR protocol. PCR primers were designed using primer-BLAST (http://www.ncbi.nlm.nih.gov/tools/primer-blast/) and 2 uM of forward and reverse primers specific to each sample were used (primers 21–38 in *Supplementary file 3*). GAPDH is a general housekeeping gene that was used as a positive control during every round of PCR.

## Conditional gene expression

### Generation of loss and gain of function MARCM clones

To induce *Toll-2* mutant MARCM clones, *w/Y; neo FRT42D, Toll-2$^{pTV}$/SM6aTM6B* recombinant flies were generated using the *neomycin resistance (neo)* selection marker. Them and *yw/Y; neo FRT42D* control male flies were crossed to *w,elavGal4 UASmCD8GFP,hs-FLP; neoFRT42D tub-Gal80$^{ts}$* virgin females, and bred at 25 ˚C. Pupae were harvested at 5 hr APF and heat shocked at 37 ˚C for 1 hr in a water-bath. The heat-shocked pupae were kept at 25 ˚C for another 90 hr. At 96 hr APF, pupal male brains were dissected, fixed and stained with rabbit anti-GFP antibodies.

To induce gain of function MARCM clones over-expressing *Toll-2* only in the adult brain, *w/Y; neo FRT42D; UAS Toll-2$^{attP2}$* and *y w/Y; neo FRT42D* control flies were crossed to *w,elavGal4 UASmCD8GFP,hs-FLP; neoFRT42D tub-Gal80$^{ts}$* female virgins, and bred at 25 ˚C. Adult flies were collected 0–3 hr post-eclosion and heat shocked at 37 ˚C for 1 hr in a water-bath. Heat-shocked flies were kept at 25 ˚C for another 24 hr and then male brains were dissected, fixed and stained with rabbit anti-GFP antibodies.

## Conditional over-expression and RNAi knock-down with *tubGAL80$^{ts}$*

To over-express or knock-down *Toll-2* in the pupa or adult brain only, we used the temperature sensitive form of the GAL4 repressor, *GAL80$^{ts}$*, ubiquitously expressed under the control of *tubulin* promoter. Flies bearing *Toll-2$^{pTV}$GAL4 UAS-histone-YFP/+; tubGAL80$^{ts}$* were used to count cell number in Kenyon cells and optic lobes, and flies bearing *MyD88GAL4 UAShistoneYFP/+; tubGAL80$^{ts}$* were used to count cells in the central brain. At 18˚C GAL80$^{ts}$ represses *GAL4* expression, whereas at 30˚C, GAL80$^{ts}$ is inactivated enabling *GAL4* expression. To conditionally manipulate *Toll-2* expression in pupa, embryo collections were left to develop at 18 ˚C, pupae were harvested 0–3 hr after puparium

formation (APF) and moved to a 30 °C incubator, where they were kept for 72 hr. Fluorescent pupae that expressed *UAS-histone-YFP* under the control of *MyD88GAL4* or *Toll-2GAL4*, were selected to be dissected, fixed and scanned.

The adult critical period for structural plasticity spans from adult day 1 to 5 (*Technau, 1984*; *Heisenberg et al., 1995*; *Barth and Heisenberg, 1997*; *Barth et al., 1997*). To conditionally manipulate *Toll-2* expression at the adult critical period and analyse the consequences in the central brain and kenyon cells, embryo collections were left to develop at 18 °C, adult progeny flies were selected and harvested 0–3 hr after eclosion, moved to a 30 °C incubator where they were kept for 48 hr, and then brains were dissected, fixed and scanned. To conditionally manipulate *Toll-2* expression at the adult critical period and analyse the consequences in the optic lobes, embryo collections were left to develop at 18 °C, adult progeny flies were selected and harvested 0–3 hr after eclosion, and kept at 18°C for another 96 hr. Fluorescent pupae were selected at 96 hr APF, moved to 30 °C for further 6 hr, followed by 24 hr at 25 °C, and then brains were dissected, fixed and scanned. For all cell counting experiments in the adult brain, only female flies were used.

## Visualisation of cell proliferation with the S-phase marker PCNA-GFP

To test whether Toll-2 can induce proliferation, we used the S-phase marker PCNA-GFP, whereby GFP is expressed under the control of the PCNA promoter containing E2F binding sites. Control (*PCNA-GFP/+;+/+;hs-Gal4/+*) or test (*PCNA-GFP/+; UAS Toll-2$^{EP709}$/+; hs-Gal4/+*) embryos were allowed to develop at 25°C, pupae were harvested at 0–1 hr APF and after further 24 hr, they were heat shocked at 37 °C for 30 min in a water bath to induce *Toll-2* expression. To allow for cell cycle onset, pupae were transferred again to 25°C for a further 9 hr, after which brains were selected and dissected, fixed and stained with anti-GFP antibodies. To test induction of proliferation in the adult brain, female flies of the same genotypes as above were selected with genetic markers, harvested 0–1 hr after eclosion, heat shocked at 37 °C for 30 min in a water bath, returned to 25 °C for another 9 hr, and then their brains were dissected, fixed and stained with anti-GFP antibodies.

## Neuronal activation

To activate neurons with temperature, we used the heat-activated cation channel TrpA1 expressed under GAL4 control. *Toll-2$^{pTV}$GAL4, UAS-histone-YFP/+; tubGAL80$^{ts}$/UAS-TrpA1 and Toll-2$^{pTV}$GAL4, UAS-histone-YFP/UAS-Toll-2RNAi; tubGAL80$^{ts}$/UAS-TrpA1* Flies were kept at 18°C from embryo until 4 days after eclosion (96 hr AE). Flies were then transferred to 30°C incubator for 5 hr to switch off Tubulin Gal80$^{ts}$ and switch on Gal4, allowing for translation. To prevent phototoxicity, flies were then shifted between a 30°C water bath and an 18°C incubator every 10 min for an hour. Finally, flies were moved to a 23°C incubator for another 24 hr and then dissected, fixed and scanned.

## Immunostainings

Immunostainings in the pupal and adult brain were carried out following standard procedures. **Primary antibodies used were**: rabbit-anti-GFP at 1:250 (Molecular Probes); rabbit anti-DsRed at 1:100 (Clontek); goat anti-Toll-1 at 1:10 (Santa Cruz); anti-FoxO at 1:500 (gift of P. Leopold); Guinea pig - anti-Dpn at 1:1000 (gift of Y.Jan); rat anti-Mira at 1:10 (abcam); mouse-anti-Repo 1:10 (Developmental Studies Hybridoma Bank, DSHB); rat-anti-Elav at 1:250 (DSHB). **Secondary antibodies used:** Alexa Donkey-anti-Rabbit 488 at 1:250; Alexa Goat-anti-Rabbit 546 at 1:250; Alexa Goat-anti Guinea-pig 633 at 1:250; Alexa Goat-anti-Rat 647 1:250; Alexa Goat-anti-mouse 647 at 1:250.

## Microscopy, imaging and development of DeadEasy plug-ins for automatic cell counting

All microscopy images were captured using laser scanning confocal microscopy with either a Zeiss LSM710 or a Leica SP8 laser scanning confocal microscopes. For image data, brains were scanned with at a resolution of either 512 × 512 or 1024 × 1024. With the Leica SP8, we used the 20x oil objective for whole adult brains and pupal brains, and zoom 0.9 for pupal brain only; 40x oil objective for optic lobes; and 63x oil objective for Kenyon cells, with step size: pupa: 0.25 µm, adult: 0.2 µm; acquisition speed was 400 Hz, airy 1, and no line averaging was applied; sections were 0.96 µm

or 1 μm apart, unless otherwise indicated. With the Zeiss LSM710, for whole adult brains, we used a 25x objective, zoom 0.6, 1 or 2 rounds of averaging, speed 7–9, and step size 1 μm.

To automatically count cell number in the adult brain, we purposely developed software plug-ins to be run with ImageJ, which we called DeadEasy Central Brain, DeadEasy Optic Lobes and Dead-Easy Kenyon Cells. These are image processing programs that identify and count cells labeled with the nuclear marker Histone-YFP in a 3D stacks of confocal images throughout the brain. Brains expressing *UAS-Histone-YFP* under the control of *GAL4*, were dissected and fixed, and without staining, scanned throughout at the confocal microscope, using settings described above. The plug-ins identify cells in the following way: (1) DeadEasy Optic Lobes: First, noise is reduced using a median filter. Then,particles whose diameter is greater than the minimum cell radius and whose brightness is above a given threshold found empirically, are retained as cells. A black majority filter is then applied to better separate cells and eliminate noise further. Finally, the very small and very large particles that are not considered cells are eliminated and the identified cells are labeled and counted. (2) DeadEasy Central Brain: In this algorithm, unlike the previous one, instead of the medium filter, the small particles in the original image are eliminated by using a minimum filter followed by a maximum filter, then the same steps as for the previous algorithm are followed. (3) DeadEasy_Mushroom_body: In this algorithm a medium 3D filter is first applied in order to eliminate noise. Then the separation between cells is accentuated, by using a minimum 3D filter followed by a maximum 3D one. Then, the regions whose radius is smaller than the radius of a cell are eliminated. To identify the cells in each region that appear too crowded, we proceed to identify recursively the spheres of variable radius that can be located in each region. It begins with spheres of greater radius, then decrease in size. Once the cells are identified they are labeled and counted. Statistical analysis was then carried out as described below. Brain size was measured using ImageJ.

## Behaviour

### Survival and longevity tests

To ask whether *Toll-2^{pTV}* mutations affected viability, we scored the relative number of homozygous mutant pupae in a population. All test stocks were balanced over *SM6aTM6B*, which results in joint second and third balancer chromosome segregation, enabling the use of Tb⁻ as a pupal marker for both second and third chromosomes. Tb⁻ can be easily seen as shorter pupae than wild-type. To score adult flies, stocks were balanced over CyO. Crosses between heterozygous mutant flies over a balancer were set, bred at 25°C and then either pupae or adult flies were scored. The survival index (S.I.) for the *Toll-2* mutations on the second chromosome was calculated as: S.I. = number of flies without balancers (Tb⁺)/balancer (Tb⁻) x 2, which for a wild-type population would be 1, if the segregation were exactly mendelian. Data were analysed using Chi-square statistical tests.

For longevity analysis, 10–20 flies of the same genotype eclosed on the same day were collected and pooled into a vial, and the day of eclosion was called Day 1. At the end of each day, flies from each vial were anaesthetized on the $CO_2$ pad and counted. All the flies that could move were scored as alive. Flies were transferred into clean vials with freshly made food once every two days.

### Negative geotaxis or climbing assay

To ask whether *Toll-2^{pTV}* mutations affected locomotion, we used the negative geotaxis or climbing assay (*Barone and Bohmann, 2013*). Flies were collected on eclosion, which was called day 1 (D1), and 20–30 D4-6 old flies were grouped into a vial per genotype, and tested as published. Essentially, vials with flies of various genotypes were tapped at once and filmed, and the number of flies of each genotype that were able to walk over a 2 cm line in 10 s was counted in the videos. The test was repeated 15 times for each genotype, separated by 30 s rest intervals, for each biological replicate, and three biological replicates were used. The experiments were carried out in a dedicated fly behaviour room with constant temperature at 25°C.

### Buridan arena tests for visual and motor system function

To test for the effect of manipulating *Toll-2* function in vision and locomotion, we used an adaptation of the classical Buridan arena test (*Strauss and Heisenberg, 1993*; N Strausfeld, method unpublished). A single fly at a time was placed in a circular arena lit up by LED-lights along its perimeter, except for two diametrically opposed dark bands, was left to walk, and filmed for 13 min. The

trajectory taken by the fly was plotted as an image, and the movements of the fly were directly recorded. The position of the fly was given at each time point, and the parameters distance travelled, time spent walking, speed, number of walking bouts, distance covered in each bout and bout duration, were extracted automatically by developing Excel Macros. The data are represented in box-plots in *Figure 3K–N*.

## Statistical analysis

Statistical analyses were carried out using Graph-Pad Prism. Confidence interval was 95%, setting significance at p<0.05. Categorical data were analysed with Chi-Square tests, followed by Bonferroni multiple comparisons corrections. Numerical data were tested first for their type of distributions. If data were distributed normally, Student t-tests were used for comparisons between two samples types or genotypes; when more than two samples were being compared, equality of variances was tested with either a Levene's or Barlett's test, and if found to be equal, One-Way ANOVA was used to test for significance, followed by a post-doc Dunnett test for multiple comparisons to control. If quantitative data were not distributed normally, then Mann-Whitney U-test was used for comparisons between two genotypes, and Kruskal-Wallis for comparisons between more, followed by post-hoc Dunn's test for multiple comparisons to control. Post-hoc Bonferroni multiple comparisons test was used to compare all samples with all samples in a given experiment.

## Acknowledgements

We thank members of our and Carolina Rezaval's lab for discussions; Lizzie Connolly for proof-reading; Richard Baines for advice on preventing excitotoxicity; Yun Fan, Iswar Hariharan, Kieran Harvey, Kenneth Irvine, Duojia Pan, Nic Tapon, Bloomington, Kyoto and Vienna Stock Centres for flies; Iowa-DSHB for antibodies; Alessandro Di Maio and BALM for technical assistance with the microscopy facility. This work was funded by a BBSRC Project Grant and BBSRC ISIS Travel Grant to AH; Marie-Curie Skłodowska Postdoctoral Fellowship to JW; BBSRC-MIBTP Studentship to NA.

## Additional information

### Funding

| Funder | Grant reference number | Author |
| --- | --- | --- |
| Biotechnology and Biological Sciences Research Council | BB/P004997/1 | Alicia Hidalgo |
| Biotechnology and Biological Sciences Research Council | BB/R017034/1 | Alicia Hidalgo |
| H2020 Marie Skłodowska-Curie Actions | NPN | Jill S Wentzell |

The funders had no role in study design, data collection and interpretation, or the decision to submit the work for publication.

### Author contributions

Guiyi Li, Data curation, Formal analysis, Investigation, Visualization, Methodology, Writing - original draft, Writing - review and editing; Manuel G Forero, Software, Methodology; Jill S Wentzell, Data curation, Formal analysis, Funding acquisition, Investigation, Visualization, Methodology; Ilgim Durmus, Ruiying Jiang, Investigation, Methodology; Reinhard Wolf, Data curation, Software, Formal analysis, Validation, Investigation, Visualization, Methodology; Niki C Anthoney, Mieczyslaw Parker, Data curation, Formal analysis, Investigation, Visualization, Methodology; Jacob Hasenauer, Investigation, Visualization, Methodology; Nicholas James Strausfeld, Methodology, Writing - review and editing; Martin Heisenberg, Conceptualization, Resources, Software, Formal analysis, Supervision, Validation, Methodology, Writing - review and editing; Alicia Hidalgo, Conceptualization, Resources, Data curation, Formal analysis, Supervision, Funding acquisition, Validation, Writing - original draft, Project administration, Writing - review and editing

## Author ORCIDs

Guiyi Li (iD) https://orcid.org/0000-0001-9620-5139
Niki C Anthoney (iD) https://orcid.org/0000-0003-3311-6328
Nicholas James Strausfeld (iD) http://orcid.org/0000-0002-1115-1774
Martin Heisenberg (iD) http://orcid.org/0000-0002-4462-8655
Alicia Hidalgo (iD) https://orcid.org/0000-0001-8041-5764

## Decision letter and Author response

Decision letter https://doi.org/10.7554/eLife.52743.sa1
Author response https://doi.org/10.7554/eLife.52743.sa2

# Additional files

## Supplementary files

• Supplementary file 1. Key Resources Table.

• Supplementary file 2. Genotypes, samples size and statistical analysis. All details of the quantitative analyses including: genotypes, sample sizes, statistical tests carried out: normality tests, parametric and non-parametric tests for two or more samples, test results and p-value for whole data sets, and multiple comparison post-hoc correction tests and their p-values.

• Supplementary file 3. List of primers. Primers used for all cloning and RT-PCR experiments.

• Transparent reporting form

## Data availability

All data generated or analysed during this study are included in the manuscript and supporting files.

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
