## [Decision Letter]

[Editors’ note: the authors submitted for reconsideration following the decision after peer review. What follows is the decision letter after the first round of review.]

Thank you for choosing to send your work, "A Toll-receptor map underlies structural brain plasticity", for consideration at *eLife*. Your article has been reviewed by three peer reviewers, and the evaluation has been overseen by a Senior Editor in consultation with a member of the Board of Reviewing Editors. Although the work is of interest, we very much regret to inform you that the findings at this stage were considered too preliminary for further consideration at *eLife*.

Specifically, numerous data sets presented here are not well controlled (Validation of several manipulations and reagents are lacking; e.g. one RNAi per gene), the data are overinterpreted, and the experiments and logic are not well described. All reviewers agreed that the manuscript may have some merit but that significantly more work will need to be done to provide a compelling story. For example reviewer sums up 12 issues, which should all be addressed. They significantly overlap with those of the other two reviewers. The link with Yki is also considered too preliminary. In summary, we suggest that you carefully read the reviewer's comments and decide if you are willing to put in a very significant amount of effort prior to resubmitting to *eLife*. All the points raised by the reviewers (except 4 of reviewer 2) will have to be addressed. Point 4 of reviewer 2 should be addressed if antibodies are available. If you decide to resubmit we expect a detailed rebuttal letter addressing all the issues that have been raised.

Reviewer #1:

This is an interesting manuscript on the role of Toll receptors in the establishment of structural brain plasticity, by a group that made seminal contribution to the characterization of the biology of receptors members from the Toll family. The background and objectives are superbly presented in the Introduction. The manuscript then presents an impressive amount of data addressing the contribution of Toll receptors to structural brain plasticity in flies. The model presented by the authors is attractive, but not entirely convincing in particular because many of the tools used lack validation and controls are missing. Below are my major comments:

1) Subsection “A Toll receptor map in the *Drosophila* brain”, last paragraph: a reference for the anti-Toll-1 antibody used and the validation of its specificity is lacking.

2) Subsection “Toll-2 is neuro-protective in the brain via MyD88”, second paragraph: what is the evidence that Myd88[NP6394] faithfully report on MyD88 expression?

3) Subsection “Toll-2 is neuro-protective in the brain via MyD88”, fifth paragraph: the fact that Toll-2 loss of function in MyD88-positive cells caused cell loss does not imply that Toll-2 maintains neuronal survival via the canonical MyD88 pathway: this is only one possible way to explain the results.

4) Subsection “Toll-2 is neuro-protective in the brain via MyD88”, sixth paragraph: the phenotype of MyD88 KO or KD flies should be compared to that of Toll-2 mutants.

5) Subsection “Toll-2 can increase cell number in development and in the adult brain”, first paragraph: correlation is not causation, the authors cannot conclude that Toll-2 maintains neuronal survival in central brain via the canonical MyD88 pathway from these experiments.

6) Subsection “Toll signaling is active in the pupal and adult brains”, title…: the claim is misleading, the data only present expression data for components of the pathway or for reporters that are not exclusive to Toll signaling.

7) Subsection “Toll signaling is active in the pupal and adult brains”, fifth paragraph: the statements regarding Dorsal and Dif are extremely confusing. What does "nuclear Dif and Dorsal" mean? Are the authors referring to the short transcript isoforms encoding proteins with NLS?

8) Subsection “Toll signaling is active in the pupal and adult brains”, fifth paragraph: experiments using transgenic flies with tagged Dif and Dorsal. NF-κB proteins are known to bind promiscuously to Rel binding sites, therefore, a loss of specificity can be expected following ectopic expression of Dif and Dorsal and should be taken into account when interpreting the results.

9) Subsection “Toll-2 can promote cell proliferation in the pupal and adult brain via Wek”, first paragraph: I do not see data in the figures showing the presence of Dorsal and Dif in the nucleus throughout pupal and adult brains.

10) Figure 4E: it is difficult to interpret the results of this experiment without knowing the phenotype that would result from single treatments (i.e. KD of JNK only, of cactus only, OE of PI3K only).

11) The Discussion is probably too long and lacks focus.

12) The figure legends are too short, often difficult to understand for the non-specialists.

Reviewer #2:

This study examines the expression patterns of Toll family receptors and their functional contribution to neurogenesis. The authors report that 1) different Toll-like receptors are expressed in distinct and overlapping patterns; 2) Toll 2 inactivation causes neurodegeneration, whereas its gain-of-function increases cell number; 3) Simultaneous knockdown of multiple Tolls profoundly reduces brain size. Based on these findings, the authors propose that distribution of diverse Toll receptors regulates structural brain plasticity in *Drosophila*.

Overall this is a rather descriptive study. Throughout the paper, the author often draw conclusions that are not supported by the actual data. The poor writing of the manuscript makes the logical progression of the paper even harder to follow. Here I just highlight a few obvious issues concerning the rigor of the studies (but there are many more in the paper).

1) In the Results section entitled "Toll-2 is neuro-protective in the brain via MyD88", the authors stated that "As MyD88 is an adaptor of Tolls, and it is required to promote neuronal survival in other contexts (Foldi, et al., 2017;Kim et al., 2007), the fact that loss of Toll-2 function in MyD88+ cells caused cell loss, implies that Toll-2 maintains neuronal survival via the canonical MyD88 pathway." Simply showing that knock-down of Toll-2 reduced the number of MyD88+ cells doesn't justify this claim. How can one conclude about the involvement of MyD88 without actually manipulating the activity of MyD88?

2) In the Results section entitled "Toll-2 can promote cell proliferation in the pupal and adult brain via Wek", the actual data that support this claim was very weak. The mere observation that Wek overexpression together with MyD88 RNAi increased cell number in the central brain alone is insufficient to draw such conclusion. How can one make such a conclusion without actually examining the requirement of Wek?

3) This study relies on a single RNAi line to investigate the function of different Toll-like receptors. At least another RNAi line should be tested to rule out the possibility of off-target effects.

4) The authors use Gal4 insertions in different Tolls as a surrogate for their expressions. Although this is a reasonable starting point, such patterns must be confirmed by orthogonal methods such as antibody staining, in situ hybridization, or at minimum a genetic rescue assay.

*Reviewer #3:*

The present manuscript by Li et al. probes the function of Toll family members in the adult *Drosophila* brain. The authors show that several Toll family members display partially overlapping expression patterns in various brain regions. Overexpression of one of these (Toll-2) induces an increase in neuronal numbers in the central brain and medulla. Concurrent depletion of several Toll family members conversely reduces cell number in the pupal brain. The authors suggest that Toll family members act via Weckle and Yki to promote adult neurogenesis by division of a population of quiescent progenitors. Although this is a potentially exciting finding that would significantly enhance our understanding of adult brain plasticity, there are several areas that require improvement before the manuscript can be accepted for publication, as detailed below.

1) Loss of function phenotypes.

Toll-2 depletion in the adult brain does not have the opposite phenotype to Toll-2 overexpression (Figure 4A). In fact, both appear to increase cell number, which is difficult to explain. The authors suggest that Tolls may act redundantly in the adult brain, which may explain why Toll-2 depletion does not reduce neuron number. They deplete several Tolls together without any temporal control, inducing pupal lethality and a reduction in brain size (Figure 7). However, since early Toll depletion elicits widespread developmental phenotypes, it is not at all clear if the results presented in Figure 7 have anything to do with the adult Toll-2 overexpression phenotypes the authors observe. The authors should deplete different Toll combinations in adult brains using tub-GAL80ts as they do in other figures and look at the effects on cell number, cell death and proliferation. If their hypothesis is correct, Toll depletion in the adult brain should compromise growth during the critical period.

2) Adult neurogenic progenitors

The authors suggest that they have identified a population of potentially proliferative MyD88+/Dpn+ adult progenitor population. In order to support this finding, they should at least directly show that these MyD88/Dpn cells can proliferation using PH3/EdU stainings in adult brains upon Toll-2 expression with the appropriate co-stainings.

3) Downstream signaling.

This part of the manuscript, in particular the connection with Yki, needs strengthening.

- Figure 6K: the control where Yki alone is being depleted is missing and should be provided. Perhaps the rescue the authors observe is an intermediate phenotype between Toll-2 expression and Yki depletion.

- To convincingly link Yki with Toll-2 signalling, the authors should test if Toll-2 overexpression can upregulate Yki reporters such as DIAP1-GFP or ex-LacZ, and/or promote Yki nuclear entry.

- In Figure 6K, the authors infer a signalling pathway where Weckle and JNK promote Yki activity downstream of Toll-2. If they wish to prove this, they should examine Yki target genes/nuclear localization upon Toll-2 expression with and without Weckle/JNK depletion.

[Editors’ note: further revisions were suggested prior to acceptance, as described below.]

Thank you for submitting your article "A Toll-receptor map underlies structural brain plasticity" for consideration by *eLife*. Your article has been reviewed by two peer reviewers, and the evaluation has been overseen by Hugo J. Bellen as the Reviewing Editor, and Utpal Banerjee as the Senior Editor. The reviewers have opted to remain anonymous. Please address the comments of the reviewers with written changes.

1) Toll-2 overexpression clearly increases cell number, but depletion of Tolls elicit complex phenotypes, making the interpretation of the data problematic. The authors still state in the Abstract that "Toll-2, loss of function caused cell death and neurodegeneration", but in my view this is misleading because, as I pointed out in my original review, these CNSs are heavily disrupted and it is not clear what causes the cell death. The authors did provide new data with temporally restricted Toll knockdown, but the results are still rather confusing, and statement such as "Signalling options could change over time, but how this comes about is not clear." are not particularly helpful. I appreciate that the authors performed many experiments to resolve this, but I would encourage them to be more cautious in their interpretations and clearly point out in the Discussion that things may not be as simple as indicated by the Toll-2 gain of function experiments would suggest.

2) Activation of Yki by Tolls. Again the authors did more experiments, but, as they point out, both the increased Yki-GFP nuclear localisation and the stg-GFP increase could simply occur as a result of increased cell cycling. In the absence of any data on non-cell cycle related Yki targets and any clear molecular mechanism connecting Yki to Toll-2, I find statements such as "Toll-2 induced cycling of adult progenitor cells via a novel Weckle and Yorkie dependent pathway" to be misleading. I would urge the authors to be more cautious and clearly state that Yki activation could be an indirect consequence of Toll-2-induced cell proliferation. They can say that a signaling pathway linking Toll-2 with Yki might exist, but that this will need further investigation in future.

---

## [Author Response]

[Editors’ note: the authors resubmitted a revised version of the paper for consideration. What follows is the authors’ response to the first round of review.]

Reviewer #1:This is an interesting manuscript on the role of Toll receptors in the establishment of structural brain plasticity, by a group that made seminal contribution to the characterization of the biology of receptors members from the Toll family. The background and objectives are superbly presented in the Introduction. The manuscript then presents an impressive amount of data addressing the contribution of Toll receptors to structural brain plasticity in flies. The model presented by the authors is attractive, but not entirely convincing in particular because many of the tools used lack validation and controls are missing. Below are my major comments:1) Subsection “A Toll receptor map in the *Drosophila* brain”, last paragraph: a reference for the anti-Toll-1 antibody used and the validation of its specificity is lacking.

Anti-Toll-1 had already been validated by others. It is a commercial antibody, from Santa Cruz Biotechnology (sc-33741): Toll (d-300). It had been previously validated by Khadilkar et al., 2017, and Lund, De Lotto and De Lotto, 2010. We have added these references within the subsection “A Toll receptor map in the *Drosophila* brain”.

2) Subsection “Toll-2 is neuro-protective in the brain via MyD88”, second paragraph: what is the evidence that Myd88[NP6394] faithfully report on MyD88 expression?

*MyD88[NP6394]* is a P-element GAL4 insertion just over 500bp upstream of the start codon of *MyD88* and importantly, within the single 5’ UTR exon (see www.flybase.org). Thus, as the GAL4 insert is within the non-intronic mRNA, it necessarily represents the endogenous expression pattern. Still, to address the reviewer’s concern, we also tried: (1) staining with a rather old and tiny aliquot we had of anti-MyD88 antibodies. Unfortunately, this did not give a robust result. (2) We generated a GFP-MyD88 CRISPR knock-in line and tested it in the adult brain. Unfortunately, this did not produce a robust GFP signal, as we made a mistake in the molecular design. (3) We bought and tested a MyD88fTG Fosmid transgenome line (Sarov et al. 2016 *eLife*), which should have expressed a MyD88-GFP fusion protein within the context of the endogenous regulatory sequences. However, this did not reveal any signal, and the authors had warned that not all their Fosmids are successful. To conclude, a lot of effort and money went into addressing this point, whilst the NP6394-GAL4 insert most likely provides the faithful profile of *MyD88* expression. We have now added an explanation on why the *MyD88[NP6394]* is appropriate in the subsection “Toll-2 is neuro-protective in the brain”.

3) Subsection “Toll-2 is neuro-protective in the brain via MyD88”, fifth paragraph: the fact that Toll-2 loss of function in MyD88-positive cells caused cell loss does not imply that Toll-2 maintains neuronal survival via the canonical MyD88 pathway: this is only one possible way to explain the results.

We had already published an exhaustive mechanistic analysis demonstrating that Toll receptors promote cell survival via MyD88 and cell death via Sarm (Foldi et al., 2017, “One of Top 10 papers of 2017”). Furthermore, this is evolutionarily conserved, as in mammals MyD88 promotes cell survival and Sarm cell death (Kim et al., 2007; Mukherjee et al., 2015). Thus, we thought it would not be necessary to provide the same type of evidence here again, nor to provide biochemical evidence that Toll-2 binds MyD88 and signals via MyD88, as we already did this for Toll-6 and -7 (Foldi et al., 2017), and MyD88 is the best-known adaptor of all Tolls and TLRs. Our data are consistent with the regulation of cell survival by Toll-2 via MyD88. Just in case the reviewer had missed this literature, we have now added sections clarifying this in the Introduction, Results and Discussion.

We showed that loss of *Toll-2* function induced neurodegeneration: it caused neuronal loss (submitted Figure 2G, H and Figure 3C-I), it reduced fly survival and lifespan (submitted Figure 3—figure supplement 1), it impaired behaviour (climbing assay and Buridan arena, submitted Figure 3—figure supplement 1C and Figure 3J-N), and caused degeneration of neurites and whole neuropiles (Figure 3C-I).

Still, to address the reviewer’s concern and ask directly whether *Toll-2* can regulate cell survival or cell death, we tested whether *Toll-2* knock-down in MyD88+ cells caused apoptosis. We stained *MyD88GAL4> UASToll-2RNAi* brains with the apoptotic marker anti-Dcp1, and counted Dcp1+ cells in 3D throughout the stack (i.e. not in projections) automatically with purposely developed DeadEasy software. Apoptotic cells get cleared away very quickly and thus can easily be missed. Therefore, we tested day 1 pupa, when most cell death occurs in normal brains, and neurons have a greater requirement to be maintained alive. We counted apoptotic cells in the central brain, as in the optic lobes there is so much cell death that DeadEasy cannot count accurately. Toll-2 knock-down in *MDy88* cells (*MyD88GAL4>UASToll-2RNAi*) increased apoptosis in the central brain. The previous data showed that *Toll-2 RNAi* knock-down caused MyD88+ cell loss, and we now also show that *MyD88 RNAi* knockdown also causes loss of MyD88+ cells (new Figure 2H). Together, these data demonstrate that *Toll-2* is required to maintain the survival of *MyD88* expressing cells.

These new data are now provided in new Figure 2G, H. We have also rewritten the text to ensure the interpretation accurately reflects the data – subsection “Toll-2 is neuro-protective in the brain”.

4) Subsection “Toll-2 is neuro-protective in the brain via MyD88”, sixth paragraph: the phenotype of MyD88 KO or KD flies should be compared to that of Toll-2 mutants.

We had already published that loss of *MyD88* function in mutants induces apoptosis and neuronal loss in the pupal ventral nerve cord (Foldi et al., 2017). Nevertheless, to respond to the reviewer’s request, we have now tested whether *MyD88-RNAi* in *MyD88* expressing neurons can induce neuronal loss in the pupal and adult brain too (*MyD88GAL4 x UAShistoneYFP, UASMyD88RNAi*), to compare to the Toll-2RNAi data in Figure 2H. We found that *MyD88-RNAi* in pupa causes cell loss in the central brain, demonstrating that the maintenance of neuronal survival by Toll-2 requires MyD88. By the adult stage, the effect of MyD88 knock-down had been restored, suggesting that MyD88 is also involved in cell proliferation, which is the focus of the second part of the paper.

These new data are now provided in Figure 2H, and changes to the text in the subsection “Toll-2 is neuro-protective in the brain”.

5) Subsection “Toll-2 can increase cell number in development and in the adult brain”, first paragraph: correlation is not causation, the authors cannot conclude that Toll-2 maintains neuronal survival in central brain via the canonical MyD88 pathway from these experiments.

This is a similar criticism to that in point (3). To respond to this reviewer’s criticism and improve the data as well as the interpretation, we have: (1) tested whether loss of *Toll-2* function in MyD88+ cells increased the number of apoptotic cells in the pupal central brain, stained with anti-Dcp1. And we have found it does. (2) Shown that *MyD88 RNAi* knock-down leads to cell loss in the pupal brain. (3) Removed Supplementary Figure 4, which are now redundant, as the data above directly demonstrate that Toll-2 and MyD88 maintain neuronal survival in pupa. These new data are now provided in new Figure 2G, H, and changes to text in the subsection “Toll-2 is neuro-protective in the brain”.

6) Subsection “Toll signaling is active in the pupal and adult brains”, title: the claim is misleading, the data only present expression data for components of the pathway or for reporters that are not exclusive to Toll signaling.

This has now been changed to “Known effectors of Toll signalling are distributed in developing and adult brains”

7) Subsection “Toll signaling is active in the pupal and adult brains”, fifth paragraph: the statements regarding Dorsal and Dif are extremely confusing. What does "nuclear Dif and Dorsal" mean? Are the authors referring to the short transcript isoforms encoding proteins with NLS?

The different isoforms of Dorsal and Dif are described in Zhou et al., 2015. Isoforms A, B, D, E and isoforms C, F in Flybase correspond to isoforms A and B, respectively, from Zhou et al., 2015. Here, the authors described isoforms Dorsal-A and Dif-A, that contain Rel-homology domain, kB site recognition in the nucleus and a nuclear localisation signal (NLS) at the C-terminus, whereas Dorsal-B and Dif-B lack the NLS. Cytoplasmic B isoforms function at the NMJ, in the recruitment of Glu receptors (Heckscher et al. 2007 Neuron). As we have previously shown that the maintenance of neuronal survival by Tolls depends on the nuclear translocation of Dif and Dorsal (Foldi et al., 2017), to function as transcription factors to drive gene expression, we were interested in visualising the localisation of the nuclear isoforms only. Thank you for the feedback, we have now added a brief explanation on this point in the subsection “Known effectors of Toll signalling are distributed in developing and adult brains”.

8) Subsection “Toll signaling is active in the pupal and adult brains”, fifth paragraph: experiments using transgenic flies with tagged Dif and Dorsal. NF-κB proteins are known to bind promiscuously to Rel binding sites, therefore, a loss of specificity can be expected following ectopic expression of Dif and Dorsal and should be taken into account when interpreting the results.

Please allow us to clarify, as it would appear that the reviewer missed the point here. Indeed, the transgenic constructs are carried in Bacmids, so the tagged proteins are expressed over wild-type levels. However, we did not investigate the phenotypic consequences of their changed levels on transcription from NFkB-binding sites on DNA. Instead, we only asked whether the tagged proteins can get into the nucleus, for which levels or binding specificity as transcription factors, are, in this context, irrelevant.

9) Subsection “Toll-2 can promote cell proliferation in the pupal and adult brain via Wek”, first paragraph: I do not see data in the figures showing the presence of Dorsal and Dif in the nucleus throughout pupal and adult brains.

It appears that reviewer 1 missed these data. These data were provided in submitted Figure 4—figure supplement 1I-L, which showed round, GFP+ cells revealing nuclear Dorsal and Dif isoforms in the pupal brain (Figure 4—figure supplement 1I, K), and nuclear Dif also in adult brain (Figure 4—figure supplement 1J). Higher magnification images were also provided (to the right of the whole brains), showing distinct, round nuclear signal.

10) Figure 4E: it is difficult to interpret the results of this experiment without knowing the phenotype that would result from single treatments (i.e. KD of JNK only, of cactus only, OE of PI3K only).

We have now carried out these single line tests for *UAS-JNK-RNAi, UAS-cactus-RNAi* and *UAS-PI3KCAAX*. Even if these had not been requested, we have also added *UAS-wek* and *UAS-MyD88RNAi* alone, for the sake of completion. We did not do this for the Kenyon Cells, as they were not significantly affected by the combined genotypes anyway. These new data are now provided in Figure 4E.

11) The Discussion is probably too long and lacks focus.

No problem, we have now shortened it, compared to the previous version. However, with the immense increase in data with the revision, this has been challenging. Still, we have made a considerable effort to keep the entire manuscript as concise as possible.

12) The figure legends are too short, often difficult to understand for the non-specialists.

No problem, we have now lengthened them. We thank this reviewer for his/her positive and enthusiastic evaluation of our work, and for the very helpful suggestions that have improved our manuscript.

Reviewer #2:[…] Overall this is a rather descriptive study. Throughout the paper, the author often draw conclusions that are not supported by the actual data. The poor writing of the manuscript makes the logical progression of the paper even harder to follow. Here I just highlight a few obvious issues concerning the rigor of the studies (but there are many more in the paper).

We are grateful to reviewer 2 for the feedback and constructive criticisms that helped us improve the manuscript.

We respectfully disagree with this reviewer that this work is ‘overall rather descriptive’ as from seven very large figures and five supplementary figures that were submitted, most data are of functional genetic analyses (e.g. cellular and behavioural phenotypes of loss and gain of function conditions and genetic epistasis tests), not descriptions. We appreciate the valid points that were raised, which we have addressed as follows:

1) In the Results section entitled "Toll-2 is neuro-protective in the brain via MyD88", the authors stated that "As MyD88 is an adaptor of Tolls, and it is required to promote neuronal survival in other contexts (Foldi, et al., 2017; Kim, et al., 2007), the fact that loss of Toll-2 function in MyD88+ cells caused cell loss, implies that Toll-2 maintains neuronal survival via the canonical MyD88 pathway." Simply showing that knock-down of Toll-2 reduced the number of MyD88+ cells doesn't justify this claim. How can one conclude about the involvement of MyD88 without actually manipulating the activity of MyD88?

We had already published an exhaustive mechanistic analysis of how Toll receptors regulate cell survival via MyD88 and cell death via Sarm (Foldi et al., 2017, selected by JCB as “One of Top 10 papers of 2017” and for the Special Collection on Cellular Neurobiology 2018” as “amongst the most exciting findings in cellular neurobiology”). Thus, we thought it would not be necessary to provide the same type of evidence here again, and we do not consider it necessary to provide biochemical evidence that Toll-2 binds MyD88 and signals via MyD88, as we already did this for Toll-6 and -7 (Foldi et al., 2017) and all Tolls can do so. We had also already published that loss of *MyD88* function in mutants induces apoptosis, as it increased the incidence of apoptotic Dcp1+ cells, and neuronal loss in the pupal ventral nerve cord (Foldi et al., 2017).

Nevertheless, to respond to the reviewer’s request, we have now manipulated the activity of MyD88, as suggested. We have tested whether *MyD88-RNAi* in MyD88 expressing neurons affects neuronal number (*MyD88GAL4 x UAShistoneYFP, UASMyD88RNAi*), to compare to the Toll-2RNAi data in Figure 2H. We found that *MyD88-RNAi* in pupa caused cell loss in the central brain. By the adult stage, the effect of MyD88 knock-down had been restored, suggesting that MyD88 is also involved in 5 cell proliferation, which is the focus of the second part of the paper. These data demonstrate that MyD88 is required to maintain cell survival in the pupal brain. These new data are provided in Figure 2H and changes to text in the subsection “Toll-2 is neuro-protective in the brain”.

To provide further evidence that solves reviewer 2’s concern, we also tested whether Toll-2 loss of function in MyD88+ cells caused apoptosis, which would lead to cell loss. We stained *MyD88GAL4xUASToll-2RNAi* brains with the apoptotic marker anti-Dcp1, followed by automatic cell counting in 3D throughout the stacks of confocal images (i.e. not in projections) with DeadEasy software we developed. Apoptotic cells get cleared away very quickly (e.g. within 20 minutes) and thus can easily be missed. Therefore, we tested day 1 pupa, as it is the time point when most cell death occurs in normal brains, and neurons have a greater requirement to be maintained alive. We counted apoptotic cells in the central brain, as in the optic lobes there is so much cell death that DeadEasy cannot count accurately. We found that Toll-2 knock-down in *MDy88* cells (*MyD88GAL4>UASToll-2RNAi*) caused an increase in apoptosis in the central brain. This demonstrates that Toll-2 is required to maintain the survival of MyD88 expressing cells. These new data are now provided in new Figure 2G.

We have also rewritten the text to ensure the interpretation accurately reflects the data – subsection “Toll-2 is neuro-protective in the brain”.

2) In the Results section entitled "Toll-2 can promote cell proliferation in the pupal and adult brain via Wek", the actual data that support this claim was very weak. The mere observation that Wek overexpression together with MyD88 RNAi increased cell number in the central brain alone is insufficient to draw such conclusion. How can one make such a conclusion without actually examining the requirement of Wek?

We agree that it is best to test also wek alone, and in fact, also MyD88-RNAi alone. Thus, in response to this reviewer, we tested:

1) whether conditional over-expression of wek alone restricted to the adult only, alters cell number (*MyD88GAL4 UAShisYFP, tubGAL80ts x UAS wek*). We show in new Figure 4E that it increased cell number compared to controls.

2) whether conditional MyD88 knock-down restricted to the adult only, affects cell number (*MyD88GAL4 UAShisYFP, tubGAL80ts x UAS MyD88RNAi*), in the adult brain. We show in new Figure 4E that it increased cell number compared to controls.

We only tested the central brain and not KCs, as no significant effect in KCs had been detected with the combined lines. Altogether, our data show that both conditional knock-down of *MyD88* and wek over-expression in the adult brain increased cell number in the brain, and that both together result in a synergistic, further increase in cell number in the adult brain. This is consistent with MyD88 promoting quiescence, and Wek inducing proliferation by antagonising MyD88.

These new data are now provided in Figure 4E, and in the subsection “Toll-2 can promote cell proliferation in the pupal and adult brain”.

We also used epistasis to test whether wek-knockdown rescued the increase in cell number caused by Toll-2 over-expression, and it did. These new data are provided in Figure 7F, and in the subsection “Toll-2 promotes cell cycling in the brain via Yorkie”.

3) This study relies on a single RNAi line to investigate the function of different Toll-like receptors. At least another RNAi line should be tested to rule out the possibility of off-target effects.

We had used a UAS-Toll-2-RNAi line from the Bloomington stock centre: TRIP[HM05241]. To address this criticism by reviewer 2 we:

1) used a second RNAi line targeting Toll-2, from the Vienna stock centre: VDRC36205. We now show that knock-down with this line (*MyD88GAL4>UAS-Toll-2RNAi[36205]*) had the same effect as the previously used TRIP line, of causing cell loss in the central brain. We did not consider it necessary to do it also with Toll-2GAL4 in Kenyon Cells, as no manipulation by any one single gene caused an effect there anyway. These data confirm that Toll-2 knock-down causes cell loss in the central brain, confirming that Toll-2 is required to maintain neuronal survival.

These new data are provided in Figure 2H and changes to text in the subsection “Toll-2 is neuro-protective in the brain”.

2) We knocked-down multiple Tolls with other RNAi lines to Toll-1, -2, -6, and -7, for Figure 7, now called Figure 8. Previously we had used lines: Bloomington stocks: UAS-Toll-1RNAi: y[1]v[1]; UAS Toll-1- RNAi [P.TRiP.JF10491]; UAS-Toll-2RNAi: y[1] sc* v[1]; P{TRiP.HM05241}attP2/TM3, Sb1; and VDRC36205. UAS-Toll-6RNAi:y[1] v[1]; P{TRiP.HMS04251}attP2; UAS-Toll-7RNAi: y sc v; P{TRIP.HM05230}attP2; UAS-Toll-8RNAi: yv; UAS Toll-8-RNAi[P.TRiP.HM05005].

Now we have also used lines: UAS Toll-1 RNAi [TRiP JF01276](BSC), UAS Toll-2 RNAi[V36305](VDRC), UAS Toll-2 RNAi [V44386] (VDRC), w; UAS Toll-6 RNAi[V928](VDRC), w;;UAS Toll-7 RNAi (NIG III)(KDRC), UAS Toll-8 RNAi[V27098] (VDRC).

The overall conclusions have not changed. That is, Tolls have redundant or overlapping functions in the brain and simultaneous knock-down of multiple *Tolls* impairs brain development.

These new data are provided in new Figure 8B, D, E and in the subsection “Tolls regulate brain size in development and in the adult critical period”.

4) The authors use Gal4 insertions in different Tolls as a surrogate for their expressions. Although this is a reasonable starting point, such patterns must be confirmed by orthogonal methods such as antibody staining, in situ hybridization, or at minimum a genetic rescue assay.

The reviewer is correct that visualising the endogenous mRNA or protein is the best way to visualise where a gene expressed. We used anti-Toll-1 antibodies, as these are commercially available (Santa Cruz Biotechnology (sc-33741) toll (d-300)), and have been validated by others (Khadilkar et al., 2017, and Lund, De Lotto and De Lotto, 2010).

However, getting good antibodies is not technically straightforward, it is costly, and neither of these methods provides high enough resolution to visualise and identify the expressing cells. So we are grateful to the Editor from exonerating us from responding experimentally to this criticism. Antibodies to Toll-6 and Toll-7 were generated in the past both by our group and others (McIlroy et al., 2013; Ward et al. 2015 Neuron), but these are noisy and the cellular resolution was poor, preventing the identification of the expressing cells. This could be due to the fact that there are 9 Tolls, and there could be non-specific binding of antibodies to multiple Tolls. Thus, we opted for not trying to generate more antibodies to all the Tolls, as we were concerned that the result may be unreliable, unpredictable and of limited use. GAL4 drivers have recurrently proved to be the best way (e.g. with membrane tethered reporters, and clones such as FlyBow and MCFO clones) to visualise and identify cells, and have been used to map entire circuits in both the larval and adult brains to the Transmission Electron Microscopy connectomes (Zheng et al. 2018 Cell; Eichler 2017 Nature, and others). The key point is where the GAL4 is placed. We generated GAL4 reporter driver lines that reproduce the spatial expression pattern of these genes, by:

1) RMCE from MIMIC insertions into the intronless coding region of *Toll-3, Toll-6, Toll-7*: since these genes have no introns, the resulting *GAL4* insertions necessarily drive their endogenous expression profiles.

2) Available Tollo^MD806^ flies, bearing a *GAL4* insertion just 180bp upstream of the *Toll-8* start codon within the 500bp 5’UTR of *Toll-8*. As the insert is within the transcribed mRNA, this means that GAL4 will necessarily drive the endogenous expression pattern.

3) CRISPR/Cas9 knock-ins for *Toll-2, Toll-4* and *Toll-5*, where *GAL4* was placed immediately adjacent 5’ of the start codon. Thus, *GAL4* will be expressed like the endogenous genes.

To conclude, we have robust evidence that all Toll reporters used reproduced their endogenous spatial profiles, demonstrating that these tools are appropriate.

To respond to this reviewer, we have improved the explanation in the subsection “A Toll receptor map in the *Drosophila* brain”. We are grateful to this reviewer for the constructive criticisms that have helped improve out work.

Reviewer #3:The present manuscript by Li et al. probes the function of Toll family members in the adult *Drosophila* brain. The authors show that several Toll family members display partially overlapping expression patterns in various brain regions. Overexpression of one of these (Toll-2) induces an increase in neuronal numbers in the central brain and medulla. Concurrent depletion of several Toll family members conversely reduces cell number in the pupal brain. The authors suggest that Toll family members act via Weckle and Yki to promote adult neurogenesis by division of a population of quiescent progenitors. Although this is a potentially exciting finding that would significantly enhance our understanding of adult brain plasticity, there are several areas that require improvement before the manuscript can be accepted for publication, as detailed below.1) Loss of function phenotypes.Toll-2 depletion in the adult brain does not have the opposite phenotype to Toll-2 overexpression (Figure 4A). In fact, both appear to increase cell number, which is difficult to explain. The authors suggest that Tolls may act redundantly in the adult brain, which may explain why Toll-2 depletion does not reduce neuron number. They deplete several Tolls together without any temporal control, inducing pupal lethality and a reduction in brain size (Figure 7). However, since early Toll depletion elicits widespread developmental phenotypes, it is not at all clear if the results presented in Figure 7 have anything to do with the adult Toll-2 overexpression phenotypes the authors observe. The authors should deplete different Toll combinations in adult brains using tub-GAL80ts as they do in other figures and look at the effects on cell number, cell death and proliferation. If their hypothesis is correct, Toll depletion in the adult brain should compromise growth during the critical period.

With due respects to this reviewer, we agree all is interesting, but please allow us to point out that these are very difficult and time consuming experiments and this manuscript has already taken 5 years of non-stop, daily, very hard work, including 5 months for this revision alone. Dealing with all those points would take much longer than appropriate for a revision.

We are grateful to this reviewer for this suggestion, which generated very interesting data. It was not technically possible to test the effect of three *Toll-RNAis* at the critical period only, due to the limitations of the genetics. So, we conditionally knocked-down two Tolls at a time (new data given in Figure 8—figure supplement 1): *Toll-2, -6* and *Toll-6,-7*. With *tubGAL80^ts^, MyD88GAL4>hisYFP, Toll-2,Toll6RNAi* this resulted in increased cell number in central brain. Toll-6 can be pro-apoptotic (Foldi et al., 2017), which would contribute to this genotype. Knock-down of Toll-6, -7 with *tubGAL80^ts^, Toll2pTVGAL4> Toll-6,Toll-7RNAi* did not affect brain size, but *Toll-2,Toll-6RNAi* knock-down did: most brains were smaller than controls, but 25% were bigger. The variability of these phenotypes suggest underlying complex compensatory interactions between multiple Tolls. Kenyon Cells, which had resisted multiple manipulations of Toll-2, were affected by knock-down of multiple Tolls. Testing Kenyon Cells further was important, as Heisenberg, Barth and Technau had previously shown that mushroom bodies exhibit experience-dependent structural plasticity. Thus, we tested tubGAL80ts, Toll2GAL4>UAShistoneYFP, UAS-Toll-2RNAi, UAS-Toll-6RNAi and UAS-Toll-7RNAi, UAS-Toll-6RNAi doubles in a Toll-2 heterozygous mutant background (Toll-2GAL4 is mutant). This showed that knock-down of multiple Tolls at the adult critical period only disorganised KC clusters, which were deeper along the A/P axis, and Kenyon Cell number. Interestingly, *Toll-6,-7* knock-down resulted in an increase in KCs. These data show that Tolls have redundant functions in KCs also at the adult critical period, and that different Tolls influence KCs in different ways. These new data are shown in new Figure 8F, G, H and in the subsection “Tolls regulate brain size in development and in the adult critical period”.

Following from this reviewer’s insight, we also tested whether conditional gain of function of *Toll-2* and its downstream effectors at the adult critical period only affected brain size. And we found that it did: over-expression of *Toll-2, MyD88RNAi* or *wek* increased brain size in the adult. These very remarkable results provided robust evidence that the Toll signalling system regulates structural brain plasticity in the adult. These new data are provided in Figure 8I,K.

We cannot explain all phenotypes and all the variability encountered, but this is due to a very clear message. Tolls regulate multiple signalling pathways that modulate cell survival, death, quiescence and proliferation. Each Toll has differential preferences for these pathways. Furthermore, as their expression patterns overlap, and loss of function in one Toll could affect how others respond, the range of potential outcomes is very complex. See Discussion.

2) Adult neurogenic progenitorsThe authors suggest that they have identified a population of potentially proliferative MyD88+/Dpn+ adult progenitor population. In order to support this finding, they should at least directly show that these MyD88/Dpn cells can proliferation using PH3/EdU stainings in adult brains upon Toll-2 expression with the appropriate co-stainings.

Adult progenitors had been previously discovered by the Ito (Kato et al., 2009), Hariharan (Siegrist et al., 2010), Moreno (Fernandez-Hernandez et al. 2013 Current Biology) and Cohen (Foo et al., 2017) labs. Our data are consistent with the above, and show that adult progenitors are mostly kept quiescent via MyD88 signalling, and Toll-2 gain of function via Wek can override MyD88, and make them divide.

EdU/BrdU reveals cells in S-phase, and we had already shown that over-expression of Toll-2 in the adult only, with heat-shock GAL4, increased the number of PCNA-GFP+ cells in S-phase. To improve the presentation of these data, we have now added PCNA-GFP+ cell counting, these data are provided in new Figure 5E. This shows that over-expression of Toll-2 in adult brain increases the incidence of cell cycling from G1 to S phase.

pH3 reveals cells in mitosis, which is very brief and hence difficult to detect. For the revision, we tried anti-pH3, but did not obtain robust data. We will also attempted to generate Toll-2 gain of function Twin-Spot MARCM clones, which, contrary to the standard MRCM clones provided in Figure 5B, C, would also reveal the mother progenitor cell. However, this method did not work as well as anticipated, as the GFP/RFP-RNAis did not switch off the reporters’ expression, and the resulting clones had mixed signals. So these results had to be discarded. To overcome these challenges, we followed two further alternative approaches:

i) We used Stg-GFP fusion protein to visualize cells in G2/M transition: String (Stg) is activated by CyclinB and Yorkie to initiate the G2-M transition and entry into mitosis. Thus the Stg-GFP fusion protein reveals cells that are in G2 and about to divide. We found many Stg-GFP+ cells both in control adult brains and those over-expressing Toll-2. These new data are provided in new Figure 5I and in the subsection “Toll-2 can promote cell proliferation in the pupal and adult brain”.

ii) We used Fly-FUCCI, which labels only cycling cells. FUCCI reveals cells in G1, G1/S, G2 and G2/M phases, but not cells in G0 (i.e. that exited the cell cycle). It drives expression of nuclear degron fusion proteins to cell cycle factors E2F and cyclin-B. Thus, degronE2F-GFP is visualized in green at G1, G2 and M, and is destroyed when cells enter S phase; and degron-CyclinB-RFP is visualised in red at S, G2 and M and it is destroyed when cells enter G1. Thus, loss of GFP signal/cells indicates G1 to S transition, loss of RFP signal/cells indicates M to G1 transition (i.e. cells just exited mitosis). In 9 combination, cells that are Red+Green are in G2 or M, cells that are only Green are either just exiting cell division or in G1, and cells that are only Red are in S phase. We over-expressed Toll-2 and FUCCI in the adult only, using *Toll-2 GAL4, tubGAL80ts x UAS-FUCCI* compared with *UAS-FUCCI, UAS-Toll-2*, and analysed the cells in G1 (green), S (red), G2 and M (green and red) phases. There were cells in G1, S and G2 phases in control adult brains. Over-expression of Toll-2 resulted in increases in the number of cells in each of these phases, meaning that Toll-2 induces cell cycling. These new data are provided in new Figure 6A-D and in the subsection “Toll-2 can promote cell proliferation in the pupal and adult brain”.

iii) We visualized MyD88+ cells with FUCCI and anti-Dpn. However, since we had already shown that most of the Dpn+ progenitors in the adult brain were MyD88+, we stained brains with both FUCCI and Dpn. We found that upon over-expression of Toll-2 most Dpn+ cells were in G1 (i.e. GFP+RFP—), meaning they had either just divided or were quiescent in G1, and some were either in S-phase or G2/M. These new data are provided in new Figure 6K and in the aforementioned subsection.

3) Downstream signaling.This part of the manuscript, in particular the connection with Yki, needs strengthening.- Figure 6K: the control where Yki alone is being depleted is missing and should be provided. Perhaps the rescue the authors observe is an intermediate phenotype between Toll-2 expression and Yki depletion.

We agree, we now provide these new data (*MyD88GAL4, UAShisYFP, tubGAL80ts x UAS-yki-RNAi*) in new Figure 7E, E’ and in the subsection “Toll-2 promotes cell cycling in the brain via Yorkie”.

- To convincingly link Yki with Toll-2 signalling, the authors should test if Toll-2 overexpression can upregulate Yki reporters such as DIAP1-GFP or ex-LacZ, and/or promote Yki nuclear entry.

We attempted the following approaches:

a) We tested whether Toll-2 promotes the expression of Yki target gene reporters ex-lacZ and diap1-lacZ (which is closer to endogenous diap1 expression than *diap1-GFP*), in *Toll-2GAL4, tubGAL80ts x ex-lacZ; UASToll-2 and Toll-2GAL4, tubGAL80ts x UASToll-2; diap1-lacZ*, with anti-βgal stainings. However, there was tremendous lacZ perdurance, and virtually all cells in the brain were βgal+, also in controls. We tried several protocols to attempt to destroy lacZ with heat, but this did not work. So this experiment was technically unfeasible.

Instead, we visualised Stg-GFP, which is also a target of Yorkie – both in controls and upon over-expression of Toll-2. These new data are provided in new Figure 5I and in the subsection “Toll-2 can promote cell proliferation in the pupal and adult brain”.

b) We tested whether Toll-2 promotes nuclear entry of Yki visualised with anti-Yki antibodies in *Toll-2GAL4 tubGAL80ts x UASToll-2* and wild-type control brains. We requested anti-Yki antibodies from the Duojia Pan, Kieran Harvey, Iswar Hariharan, Nic Tapon and Kenneth Irvine labs, and we obtained two anti-Yki aliquots, independently generated by the Duojia Pan and Ken Irvine. Both antibodies were tested in larval and pupal CNS, as positive controls. Unfortunately, in our hands, the nuclear signal was not good enough relative to the background/cytoplasmic in positive controls to provide robust data for the CNS. Thus, this experiment was technically unfeasible too.

c) We tested whether Toll-2 “promotes nuclear entry of Yki-GFP” in *Toll-2GAL4, tubGAL80ts x Yki-GFP, UASToll-2* brains. For this, we obtained two independently generated Yki-GFP fusion protein stocks, from Nic Tapon and Harvey Kiran. This worked well. We found 10 abundant Yki-GFP+ cells in the normal brain, in controls. We also found a change in the distribution of in Yki-GFP+ cells in brains with Toll-2 over-expression. These cells were difficult to count throughout the brain partly because there were many cells, and partly because automatic counting with DeadEasy was not possible due to broad regions with cytoplasmic signal. Thus, we counted Yki-GFP+ cells manually in three distinct Regions of Interest (ROIs): the medulla of the optic lobe, the sub-esophageal ganglion (SOG) and an anterior, top-left corner of the central brain. The data showed that: (1) there are cells in G2- M in the normal adult brain; (2) and that over-expression of Toll-2 caused either a bimodal distribution – some brains with fewer cells, some brains with more cells than controls – or an increase in cells with nuclear Yki. Yki trafficking is highly dynamic, and cells shuttle in and out of the nucleus (see Manning et al., 2018. As cells cycle, and transition between phases (e.g. G2-M to G1, or G0), they can translocate Yki out of the nucleus, or even down-regulate it. Thus, these data indicate that Toll-2 stimulates cell cycling. These new data are provided in new Figure 7A-D, and in the subsection “Toll-2 promotes cell cycling in the brain via Yorkie”.

- In Figure 6K, the authors infer a signalling pathway where Weckle and JNK promote Yki activity downstream of Toll-2. If they wish to prove this, they should examine Yki target genes/nuclear localization upon Toll-2 expression with and without Weckle/JNK depletion.

We have previously published that Wek functions downstream of Tolls and upstream of JNK in the context of promoting apoptosis (Foldi et al., 2017), and others have previously published that JNK functions upstream of Yki (Katsukawa et al., 2018). We have also previously published that Wek and MyD88 are the key intracellular Toll adaptors that can swing between distinct cellular outcomes downstream of Tolls (Foldi et al., 2017). Thus, the key gene to test is wek, rather than JNK.

The epistasis experiments suggested by reviewer 3 would have been: *MyD88GAL4, tubGAL80ts* flies crossed to *UAS Toll-2, UAS wek-RNAi* and visualising nuclear Yki with anti-Yki antibodies, plus controls. We tried this. However, the anti-Yki antibodies did not work, see above, so this could not be done. Introducing Yki-GFP into the genotypes above would have required the generation of recombinants plus 2^nd^ and 3^rd^ chromosome combinations, plus then the experiment requires breeding at 18°C, rendering the genetics for this experiment alone 6 months long – plus the time required to do the experiment. This was disproportionate for a revision.

As an alternative approach, we tested whether, like yki-RNAi, knock-down of wek also rescues the increase in cell number caused by Toll-2 gain of function: *MyD88GAL4 UAShisYFP, tubGAL80ts x UAS Toll-2, UAS wek-RNAi*. And indeed, it does: knock-down of either yki or wek rescued the increase in cell number caused by Toll-2 over-expression. These data showed that both Yki and Wek function downstream of Toll-2 to regulate cell number in the adult critical period. These new data are provided in new Figure 7E, E’ and in the subsection “Toll-2 promotes cell cycling in the brain via Yorkie”.

[Editors’ note: what follows is the authors’ response to the second round of review.]

[…] 1) Toll-2 overexpression clearly increases cell number, but depletion of Tolls elicit complex phenotypes, making the interpretation of the data problematic. The authors still state in the Abstract that "Toll-2, loss of function caused cell death and neurodegeneration", but in my view this is misleading because, as I pointed out in my original review, these CNSs are heavily disrupted and it is not clear what causes the cell death. The authors did provide new data with temporally restricted Toll knockdown, but the results are still rather confusing, and statement such as "Signalling options could change over time, but how this comes about is not clear." are not particularly helpful. I appreciate that the authors performed many experiments to resolve this, but I would encourage them to be more cautious in their interpretations and clearly point out in the Discussion that things may not be as simple as indicated by the Toll-2 gain of function experiments would suggest.

We thank the reviewers for their feedback, as this helps us improve our manuscript. We have now revised the text, following the recommendation to be more cautious in the interpretations, and highlight the complexity of Toll signalling outcomes, in the Discussion.

Please allow me to discuss that there are separate issues in the statement above.

1) “depletion of Tolls elicit complex phenotypes, making the interpretation of the data problematic”…. “these CNSs are heavily disrupted and it is not clear what causes the cell death”.

This sentence is ambiguous, and we wonder if the reviewer refers to the knock-down of multiple *Tolls* in Figure 8, which reduced brain size. The phenotype of the multiple *Toll* knock-down most likely results from reduced cell survival and reduced cell proliferation, throughout development. We provide increased evidence that Toll-2 promotes cell proliferation (e.g. new Figure 6 and 7). The point on cell death is dealt with below.

To clarify the points made on the phenotype multiple *Toll* knock-down, we have revised the text in the Results, and rewritten the Discussion.

2) “The authors still state in the Abstract that "Toll-2, loss of function caused cell death and neurodegeneration", but in my view this is misleading because, as I pointed out in my original review, these CNSs are heavily disrupted and it is not clear what causes the cell death.”

The reviewer doubts the interpretation that *Toll-2* loss of function caused cell death and neurodegeneration because “depletion of Tolls” results in “heavily disrupted and it is not clear what causes the cell death”.

In response to the reviewers, for the revised version we provided evidence that loss of *Toll-2* function induces apoptosis which was visualised with Dcp1 (Figure 2G). This demonstrated that loss of Toll-2 causes cell death. We also provided further evidence in the revised version that *Toll-2* knock-down using multiple RNAis and also *MyD88-RNAi* knockdown both caused cell loss (Figure 2H) in fact, loss of MyD88+ cells. Thus, loss *Toll-2* function via MyD88 induced apoptosis and cell loss. We had already published an exhaustive mechanistic analysis demonstrating that Toll receptors promote cell survival via MyD88 and cell death via Sarm (Foldi et al., 2017 “One of Top 10 papers of 2017”). Furthermore, this is evolutionarily conserved, as in mammals MyD88 promotes cell survival and Sarm cell death (Kim et al., 2007; Mukherjee et al., 2015). In the revised version, we provided new evidence that enhancing the pro-survival MyD88 pathway by knocking-down the NFκB inhibitor Cactus, or pro-apoptotic JNK, both increased cell number (Figure 4E). This demonstrated that Toll(s) signalling via the MyD88 pathway maintains neuronal survival in the brain.

We also showed that Toll-2 is not pro-apoptotic, as over-expression of *Toll-2* did not decrease cell number.

We also showed that Toll-2 caused neurodegeneration. *Toll-2* mutant MARCM clones resulted not only in massive cell loss, but also in axonal atrophy and misrouting, and dendrite loss in remaining cells (Figure 3A-G). We showed that *Toll-2* loss of function mutants have reduced longevity, impaired climbing and impaired locomotion – standard behavioural evidence for neurodegeneration used by the *Drosophila* community.

Thus, we demonstrated that *Toll-2* loss of function induced apoptosis, caused neuronal loss, caused neurite atrophy and misrouting, impaired longevity and behaviour. These are evidence that *Toll-2* loss of function causes neurodegeneration. We demonstrated that the neurons lost were MyD88+, that *MyD88* loss of function also caused loss of MyD88+ neurons, and that neuronal number could be rescued by enabling MyD88 signalling. Thus, this demonstrated that the mechanism underlying the regulation of cell survival in the brain was the same as described before for the pupa (see Foldi et al., 2017).

In response to this reviewer, to clarify the points made on cell survival, cell death and neurodegeneration, we have revised the text in Results, and rewritten the Discussion.

3) “The authors did provide new data with temporally restricted Toll knockdown, but the results are still rather confusing, and statement such as "Signalling options could change over time, but how this comes about is not clear." are not particularly helpful….. clearly point out in the Discussion that things may not be as simple as indicated by the Toll-2 gain of function experiments would suggest.”

Indeed, we had provided evidence that conditional knock-down of *Toll-2* in adult could induce a compensatory response resulting in a compound phenotype, and that Tolls also have redundant functions, as knock-down of *Toll-2* alone did not affect Kenyon cells, whereas knock-down of 2 or more *Tolls* did. For instance, in the original version we stated “So far, data were consistent with Toll-2 maintaining neuronal survival via the canonical MyD88 pathway in the brain. However, multiple Tolls can regulate this pathway, and Tolls can also promote apoptosis via non-canonical signalling pathways (Foldi et al., 2017), thus altering the levels of Toll-2 could cause compensation by other Tolls, compounding the phenotypes.” In the revised version, we provided new evidence that both redundancy and compensation contribute to phenotypes, as conditional knock-down of two *Tolls* at a time altered cell number, but phenotypes differed with different combinations of Tolls (Figure 8F). This meant that each Toll can elicit distinct outcomes (as we proposed in Foldi et al., 2017), and knock-down of one *Toll* can induce a compensatory response by another *Toll*, and not necessarily in the same direction (Figure 8F). Indeed, signalling by the Toll system is complex.

To clarify the points on redundancy and compensation, we have now clarified the text in Results, and Discussion.

To conclude, to address the concerns of this reviewer, we have: (1) revised the Abstract; (2) revised the text in Results and Discussion.

We thank this reviewer for their feedback and we hope they are satisfied with our improvements to the manuscript.

*2)* “Activation of Yki by Tolls. Again the authors did more experiments, but, as they point out, both the increased Yki-GFP nuclear localisation and the stg-GFP increase could simply occur as a result of increased cell cycling. In the absence of any data on non-cell cycle related Yki targets and any clear molecular mechanism connecting Yki to Toll-2, I find statements such as "Toll-2 induced cycling of adult progenitor cells via a novel Weckle and Yorkie dependent pathway" to be misleading. I would urge the authors to be more cautious and clearly state that Yki activation could be an indirect consequence of Toll-2-induced cell proliferation. They can say that a signaling pathway linking Toll-2 with Yki might exist, but that this will need further investigation in future.”

The reviewer argues that the new Yki-GFP data provided for this revised version – together with the previously provided Stg-GFP data – are evidence of cell proliferation but not necessarily as a result of Toll-2 directly activating Yki via Wek. As Stg is required for cell proliferation, and it is a target of Yki, presumably the reviewer means that the trigger for cell proliferation induced by Toll-2 may reside upstream. Please allow me to comment on the points above.

We are delighted that this reviewer considers that the Stg (target of Yki) and new Yki data (Figure 7) are satisfactory evidence of cell cycling induced by Toll-2.

A direct link of Toll signalling to Yki had already been published by others, both in the context of immunity and cell competition (Liu et al., 2016; Katsukawa et al., 2018). Thus, it would be reasonable to think that Toll-2 could in some contexts also use the same pathways as Toll-1.

We agree with this reviewer that our data do not enable us to state that Toll-2 activates Wek which activates Yki, and that these three factors are linked in a lineal pathway. However, we had provided genetic epistasis evidence that Yki functions downstream of Toll-2, as *yki-RNAi* knock-down rescued the increase in cell number caused by *Toll-2* over-expression. In the revised version, we provided new evidence that Wek also functions downstream of Toll-2, as *wek-RNAi* knock-down rescued the increase in cell number caused by *Toll-2* over-expression. What we did not test or show, is whether Wek is upstream of Yki and whether Toll-2, Wek and Yki are lineally related in the same pathway.

Thus, to address this criticism by this reviewer and improve the interpretation of our data, whilst representing the evidence accurately, we have now: (1) revised the Abstract; (2) revised the text in Results and Discussion; (3) and revised Figure 9C, where we now show the three possible alternatives: that Wek induces proliferation independently of Yki; that Wek activates Yki by a yet unknown mechanism; or that Wek activates Yki via JNK, as the link of Wek to JNK has been previously reported (Foldi et al., 2017) and the activation of Yki by JNK has also been reported (Katsukawa et al., 2018).